# HSV-1 and influenza infection induce linear and circular splicing of the long NEAT1 isoform

**Marie-Sophie Friedl[1], Lara Djakovic[2], Michael Kluge[1], Thomas Hennig[2], Adam W. Whisnant[2], Simone Backes[2], Lars Dölken[2], Caroline C. Friedel[1]\***

**1** Institute of Informatics, Ludwig-Maximilians-Universität München, Munich, Germany, **2** Institute for Virology and Immunobiology, Julius-Maximilians-University Würzburg, Würzburg, Germany

\* caroline.friedel@bio.ifi.lmu.de

ⓞ OPEN ACCESS

**Data Availability Statement:** All RNA-seq data analyzed in this study were downloaded from the SRA. SRA project IDs: HSV-1 WT infection time-course: SRP044766; HSV-1 deltavhs infection

## Abstract

The herpes simplex virus 1 (HSV-1) virion host shut-off (*vhs*) protein cleaves both cellular and viral mRNAs by a translation-initiation-dependent mechanism, which should spare circular RNAs (circRNAs). Here, we show that *vhs*-mediated degradation of linear mRNAs leads to an enrichment of circRNAs relative to linear mRNAs during HSV-1 infection. This was also observed in influenza A virus (IAV) infection, likely due to degradation of linear host mRNAs mediated by the IAV PA-X protein and cap-snatching RNA-dependent RNA polymerase. For most circRNAs, enrichment was not due to increased circRNA synthesis but due to a general loss of linear RNAs. In contrast, biogenesis of a circRNA originating from the long isoform (NEAT1_2) of the nuclear paraspeckle assembly transcript 1 (NEAT1) was induced both in HSV-1 infection–in a *vhs*-independent manner–and in IAV infection. This was associated with induction of novel linear splicing of NEAT1_2 both within and downstream of the circRNA. NEAT1_2 forms a scaffold for paraspeckles, nuclear bodies located in the interchromatin space, must likely remain unspliced for paraspeckle assembly and is up-regulated in HSV-1 and IAV infection. We show that NEAT1_2 splicing and up-regulation can be induced by ectopic co-expression of the HSV-1 immediate-early proteins ICP22 and ICP27, potentially linking increased expression and splicing of NEAT1_2. To identify other conditions with NEAT1_2 splicing, we performed a large-scale screen of published RNA-seq data. This uncovered both induction of NEAT1_2 splicing and poly(A) read-through similar to HSV-1 and IAV infection in cancer cells upon inhibition or knockdown of CDK7 or the MED1 subunit of the Mediator complex phosphorylated by CDK7. In summary, our study reveals induction of novel circular and linear NEAT1_2 splicing isoforms as a common characteristic of HSV-1 and IAV infection and highlights a potential role of CDK7 in HSV-1 or IAV infection.

## Introduction

Herpes simplex virus 1 (HSV-1) is one of nine herpesviruses known to infect humans [1, 2]. It is most commonly known for causing cold sores but can also result in life-threatening diseases.

time-course: SRP192356; RNA-seq of subcellular fractions: SRP110623, SRP189489, SRP191795 (same experiment but data for mutant viruses submitted separately); T-HFs-ICP22 and T-HFs-ICP22/ICP27 cells, WT strain F and deltaICP22 infection: SRP340110; WT strain KOS, deltaICP27 and ICP27 overexpression: SRP189262; IAV infection time-courses: SRP091886, SRP103821; platelets: ERP003815; erythrocytes: SRP050333; CDK7 inhibition/knockdown: VCaP, LNCaP and DU145 cells: SRP179971; HCV-29 cells: SRP217721; TE7 and KYSE510 cells: SRP068450; C666-1, HK1 and HNE1 cells: SRP101458; BxPC3, MiaPaCa-2 and PANC1 cells: SRP165924; UM-Chor1 and CH22 cells: SRP166943, SRP270819; Nalm6 cells: SRP307127; DRB treatment of HEK293 cells: SRP055770. RNase R treatment: SRP197110, SRP152310.

**Funding:** This work was supported by the Deutsche Forschungsgemeinschaft (www.dfg.de) grants FR2938/10-1 to C.C.F. and LD1275/6-1 to L.D. and by the European Research Council (erc.europa.eu) grant ERC-2016-CoG 721016 – HERPES to L.D. The funders had no role in study design, data collection and analysis, decision to publish, or preparation of the manuscript.

**Competing interests:** The authors have declared that no competing interests exist.

A key role in HSV-1 lytic infections is played by the virion host shut-off (*vhs*) protein, which is delivered to the infected cell by the incoming virus particles and rapidly starts cleaving both cellular and viral mRNAs [3]. *Vhs* target recognition occurs during translation initiation and requires binding of *vhs* to components of the cap-binding complex eIF4F [4–8]. *Vhs* can also cleave circular RNAs (circRNAs), but only if they contain an internal ribosome entry site (IRES) [9]. CircRNAs form covalent RNA circles and are naturally generated during the splicing process either by "back-splicing" of exons out of their linear order (resulting in non-colinear 3′–5′ junctions) or stabilization of intron lariats with 2′–5′ junctions joining the 5′ and branchpoint nucleotides (Fig 1A) [10]. Back-splicing requires canonical splice sites and spliceosome assembly [11]. Previously considered to occur only rarely [12], recent large-scale RNA sequencing (RNA-seq) studies revealed the existence of thousands of circRNAs in eukaryotic cells [13, 14]. Due to the absence of a 5' cap or poly(A) tail and their resistance to exonucleases, circRNAs are more stable than linear RNAs [10]. While a function as miRNA sponge has been reported for a small number of circRNAs [15, 16], this is now nevertheless thought to be rare [14]. Other functions reported for individual circRNAs involve regulation of transcription, alternative splicing and translation and some may even serve as templates for protein synthesis [14]. However, for most circRNAs their function remains elusive.

Recently, Shi *et al.* reported on dysregulation of circRNAs during HSV-1 infection, with 188 circRNAs being significantly up-regulated at 48 h post infection (p.i.) [19]. Their differential analysis was performed on FPKM values (= fragments per kilobase of transcript per million fragments mapped) calculated for circRNAs from reads crossing the circular, i.e., back-splicing, junction. For FPKM normalization, circRNA read counts were divided by the total number of reads aligning to the host genome, most of which map to linear transcripts. Thus, this analysis cannot distinguish between a true up-regulation of circRNA biogenesis and an

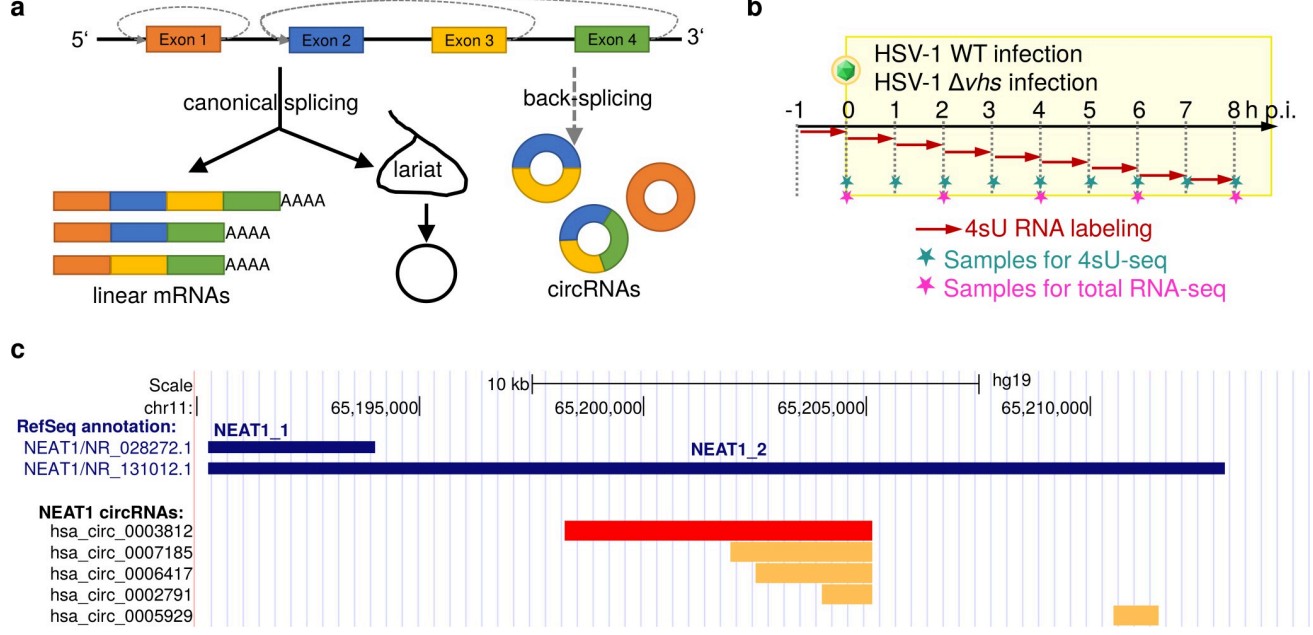

**Fig 1. Overview on circRNA biogenesis, experimental setup of HSV-1 infection time-courses, and NEAT1 transcripts and circRNAs. (a)** circRNA biogenesis occurs via back-splicing connecting the 3' end of a downstream exon with the 5' end of an upstream exon (indicated by dashed gray lines) or stabilization of lariats. CircRNAs originating from lariats are also denoted as circular intronic RNAs (ciRNAs) [17]. **(b)** Experimental setup for the total RNA- and 4sU-seq time-courses in WT and Δ*vhs* HSV-1 infection. Fig adapted from [18] (CC BY 4.0 license, © the authors). Two biological replicates were obtained for each time-point. **(c)** Genomic coordinates of the NEAT1_1 (polyadenylated) and NEAT1_2 (non-polyadenylated, stabilized by triple helix) transcripts (blue) and NEAT1_2 circRNAs observed in HSV-1 or influenza A virus (IAV) infection. The hsa_circ_0003812 circRNA induced strongly in HSV-1 and IAV infection is shown in red, while other much less abundant NEAT1_2 circRNAs are indicated in orange.

enrichment of circRNAs in case of a selective loss of linear RNAs. Considering that *vhs* cleaves translated linear transcripts but not circRNAs without an IRES, we hypothesized that the seeming up-regulation of many circRNAs in HSV-1 infection largely represented the escape of circRNAs from *vhs*-mediated RNA decay. This would result in an enrichment of circRNAs similar to, but likely less pronounced than RNase R treatment commonly used to enrich circRNAs [20]. We previously performed 4-thiouridine-(4sU)-tagging followed by sequencing (4sU-seq) to characterize *de novo* transcription in hourly intervals during the first 8 h of lytic wild-type (WT) HSV-1 strain 17 infection of primary human foreskin fibroblasts (HFFs) and combined this with total RNA-seq in two-hourly steps [21, 22] (Fig 1B, n = 2 replicates). More recently, we applied the same experimental set-up to Δ*vhs* infection to study *vhs*-dependent gene regulation [18] (Fig 1B, n = 2 replicates). In this study, we explore these data to show that *vhs*-mediated degradation of linear RNAs in HSV-1 infection leads to a general enrichment of circRNAs relative to linear circRNAs. This not only confounds analysis of differential circRNA expression but also standard differential gene expression and exon usage analyses.

Our analysis also revealed actual up-regulation of both circular and novel linear splicing of the long non-coding (linc)RNA nuclear paraspeckle assembly transcript 1 (NEAT1) during HSV-1 infection. In contrast to other circRNAs, this induction was independent of *vhs* and resulted from *de novo* synthesis of both the circRNA and linear splicing isoforms in HSV-1 infection. The nuclear lincRNA NEAT1 is one of the most highly expressed lincRNAs and has two isoforms with identical 5' but distinct 3' ends [23] (Fig 1C): A short transcript (NEAT1_1, also denoted as MENε, ~3.7nt), stabilized by a poly(A) tail and expressed in a wide range of cells [24], and a long transcript (NEAT1_2, also denoted as MENβ, ~22.7 nt), stabilized by a triple helical structure [25] but with more limited expression in particular cell types [24]. The HSV-1-induced circRNA is generated from the 3' region unique to the NEAT1_2 transcript (Fig 1C). NEAT1_2 is essential for the structure of paraspeckles [26], nuclear bodies located in the interchromatin space. NEAT1_2 forms a scaffold for paraspeckles by being bound by a number of proteins with functions in transcription and/or RNA processing and splicing (reviewed in [27, 28]). Paraspeckles impact gene expression by nuclear retention of adenosine-to-inosine-edited mRNAs [28]. In viral infection, NEAT1_2 plays a role in interleukin 8 (IL-8) induction by sequestering the IL-8 repressor splicing factor proline/glutamine-rich (SFPQ) to paraspeckles [29]. NEAT1 is up-regulated in HIV-1, dengue, HSV-1, Hantavirus, Hepatitis D and influenza infections [28]. Expression levels of NEAT1_2 have been shown to be positively correlated with presence of paraspeckles [30] and HSV-1 infection induces paraspeckle formation, likely through up-regulation of NEAT1 [31].

The HSV-1 immediate-early proteins ICP27 and ICP22 are both known to interact with the host transcription and RNA processing machinery. ICP27 regulates splicing and polyadenylation by interacting with splicing factors and the mRNA 3' processing factor CPSF [32–35]. We recently demonstrated that ICP27 is sufficient but not necessary for disruption of transcription termination in HSV-1 infection [34]. ICP22 represses RNA Polymerase II (Pol II) transcription elongation by interacting with elongation factors, such as the positive transcription elongation factor b (P-TEFb) and the FACT complex, and inhibits phosphorylation of the Pol II carboxyterminal domain (CTD) at Ser2 residues [36]. Ser2 phosphorylation plays a key role in recruiting splicing and termination factors to Pol II [37]. We thus hypothesized that ICP27 and/or ICP22 may play a role in induction of NEAT1_2 splicing. Indeed, we could show that combined ectopic expression of ICP27 and ICP22 is sufficient to induce both circular and linear splicing of NEAT1_2, however neither protein was required.

HSV-1 and influenza A (IAV) infection as well as heat stress lead to NEAT1_2 up-regulation [29, 38] and read-through transcription beyond poly(A) sites for tens-of-thousands of nucleotides for many but not all cellular genes [21, 22, 39–41]. IAV infection, but not heat

stress, also induces NEAT1_2 circular and linear splicing. Moreover, selective inhibition and knockdown of cyclin-dependent kinase 7 (CDK7) as well as knockdown of its phosphorylation target the MED1 subunit of the Mediator complex [42] induce both NEAT1_2 splicing and read-through transcription in cancer cell lines. This highlights a possible link between read-through transcription and NEAT1_2 splicing and a potential role of CDK7 in HSV-1 or IAV infection.

## Results

### *Vhs* activity leads to enrichment of circRNAs in HSV-1 infection

CircRNA detection from RNA-seq reads is based on identifying junction reads connecting a donor splice site of a downstream exon to the acceptor splice site of an upstream exon. Several algorithms for circRNA *de novo* detection are available, but due to high false positive rates combination of at least two algorithms is recommended [43]. We thus employed a pipeline combining circRNA_finder [44] and CIRI2 [45] for circRNA *de novo* detection (see methods and Fig 2A) and analyzed only circRNAs that were independently discovered by both algorithms. We applied this pipeline on our total RNA-seq and 4sU-seq time-courses for the first 8 h of lytic wild-type (WT) and Δ*vhs* HSV-1 infection (Fig 1B). This identified a total of 16,463 circRNAs in the WT time-course and 7,306 in the Δ*vhs* infection time-course, with 1,647 to 6,887 circRNAs detected per sample in total RNA (S1A Fig). In contrast, only a small number of circRNAs were detected in 4sU-RNA (119 and 194 in at least one sample of WT and Δ*vhs* 4sU-seq, respectively, S1B Fig). This is partly consistent with the study by Zhang *et al.* who also obtained smaller numbers of circRNAs (255, 820, 876 circRNAs in PA1, hESC H9 and H9 differentiated FB Neurons cells, respectively) with 1 h of 4sU-tagging using 4sUDRB-seq (DRB treatment followed by 4sU-seq after DRB removal) compared to total RNA-seq (6,740, 4,523, and 11,185 circRNAs, respectively) [46]. While this highlights a high variability between experiments and/or cell types in the number of circRNAs detected, the higher number of circRNAs detected by Zhang *et al.* can also partly explained by the use of only one circRNA detection algorithm (CIRCexplorer [47]). This is less restrictive than requiring independent identification of circRNAs by two algorithms as done in our study, but likely has a higher false positive rate. Moreover, circRNA_finder and CIRI2 require presence of canonical GT-AG splicing signals, while CIRCexplorer also identifies potential circRNAs with other less common splicing signals. Most circRNAs in our study (>97%) were only detected in total RNA and almost all of the identified circRNAs (82.6% and 92.7% for the WT and Δ*vhs* infection time-courses, respectively) were already annotated in circBase, a database of circRNAs identified in large-scale RNA-seq studies [48].

In total RNA, a substantially larger number of circRNAs was identified for WT infection than for mock and Δ*vhs* infection (S1A Fig). Since the number of reads mapping to the host genome in total RNA was comparable or higher in Δ*vhs* infection than in WT infection (S1C Fig), adjusting the read count threshold for a circRNA to be detected to sequencing depth further increased the difference in detected circRNAs between WT and Δ*vhs* infection (Fig 2A). Thus, the lower number of circRNAs identified in Δ*vhs* infection is not due to lower sequencing depth. Moreover, normalized to sequencing depth, a smaller number of circRNAs was detected in uninfected cells of the Δ*vhs* infection time-course compared to the WT infection time-course. In part, this could be due to shorter read length (76 nt in the Δ*vhs* infection time-course vs. 101 nt in the WT infection time-course, both paired-end sequencing), the effect of which will be further investigated below. However, other experimental differences between the two time-courses, which were performed at different times, likely contribute. To account for these experimental differences, we always compared samples of WT infection against mock

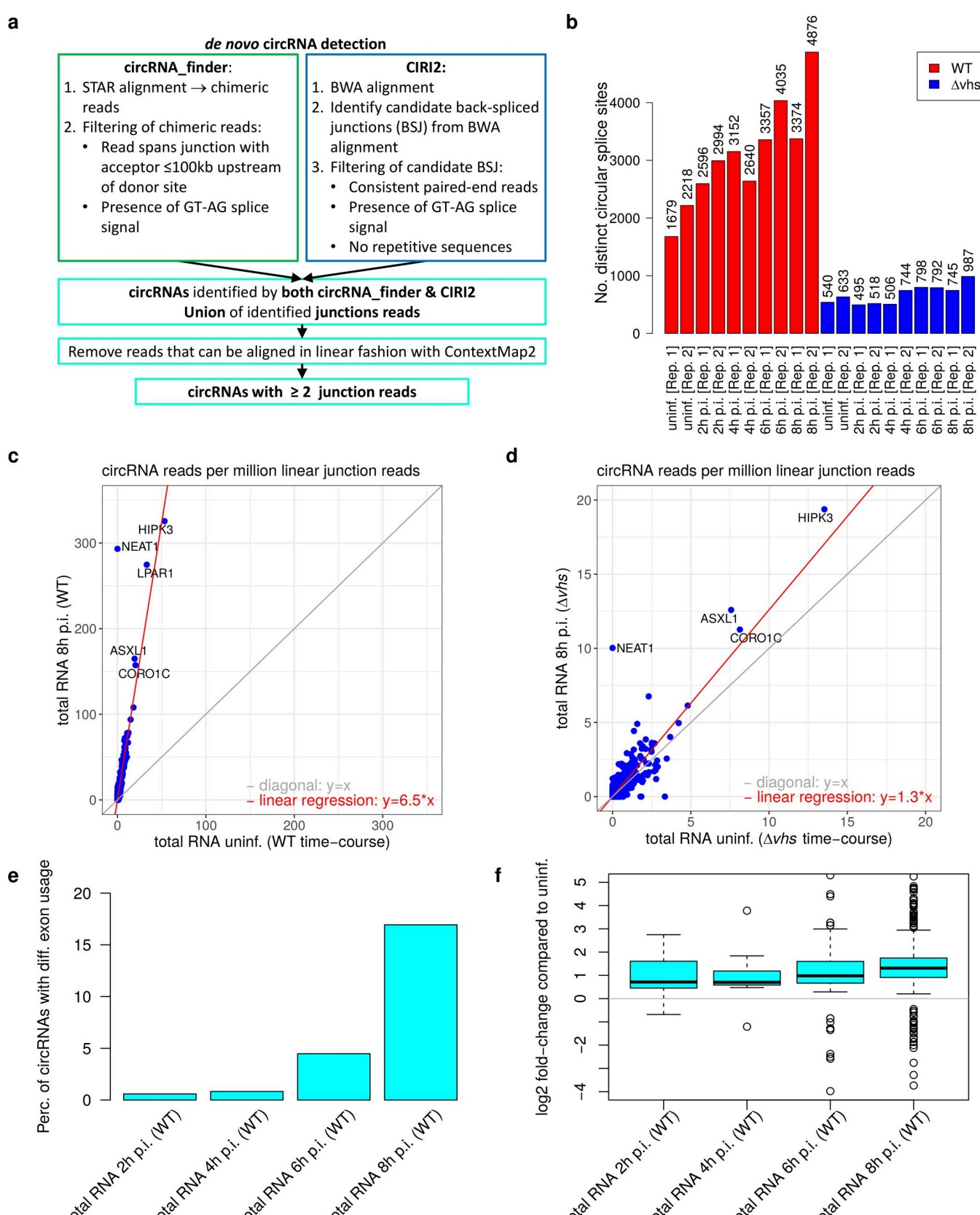

**Fig 2. *vhs*-dependent enrichment of circRNAs in HSV-1 infection. (a)** Outline of our pipeline for *de novo* circRNA detection (see also methods). It combines two different algorithms, circRNA finder [44] and CIRI2 [45], and retains only circRNAs identified independently by both algorithms. circRNA finder is based on analysis of chimeric reads determined with the RNA-seq mapper STAR [49]. Chimeric reads are reads that align to two distinct regions of the genome in manner not consistent with "normal" linear transcripts. CIRI2 is based on identifying pairs of clipped read alignments (i.e., local alignments for substrings of the read, determined with BWA [50]) for the same read where a downstream part of the read aligns upstream of an upstream part of the read (= back-spliced junction (BSJ) reads). Both methods use further filtering steps to remove false positive results. **(b)** Number of circRNAs identified for each sample of total RNA for WT and Δ*vhs* infection after adjusting the required number of reads for detecting a circRNA to the number of reads mapping to the host genome in each sample. Thus, a threshold of 2 reads was used for the sample with the lowest number of reads and higher thresholds were used for samples with higher number of host reads. Numbers of identified circRNAs with a threshold of 2 reads for each sample (as outlined in **(a)**) are shown in S1A Fig for total RNA and in S1B Fig for 4sU-RNA. **(c, d)** Scatterplots comparing normalized circRNA counts (normalized to the number of linear junction reads mapped to the host genome) between mock and 8 h post infection (p.i.) WT infection **(c)** and mock and 8 h p.i. Δ*vhs* infection **(d)**. Linear regression analysis across all circRNAs (red line) was used to estimate the enrichment of circRNAs relative to linear mRNAs in HSV-1 infection compared to mock infection. The regression estimate for the enrichment is shown on the bottom right. The gray line indicates the diagonal, i.e., equal values on the x- and y-axis. The five most highly expressed circRNAs are marked by name. Corresponding scatterplots for 2, 4 and 6 h p.i. are shown in S2 Fig. **(e)** Percentage of expressed circRNAs (= normalized circRNA count >0 in uninfected cells) at each time-point of WT HSV-1 infection for which at least one exon within the genomic region of the circRNA shows differential exon usage for the corresponding gene (determined with DEXSeq, multiple testing adjusted p-value ≤0.005). **(f)** Boxplots showing the distribution of log2 fold-changes for exons located within circRNAs. For each circRNA, only the exon with the maximum absolute log2 fold-change is shown.

infection of the WT infection time-course and samples of Δ*vhs* infection against mock infection of the Δ*vhs* infection time-course.

To evaluate enrichment of individual circRNAs relative to linear transcripts in HSV-1 infection, circRNA read counts were normalized to the total number of reads mapping to linear exon-exon junctions of all human protein-coding genes (not only the "parent" genes from which circRNAs originate) and then averaged between replicates. The reason we did not normalize only against reads from "parent" genes is that reads that map linearly to splicing junctions within circRNA regions could originate from either linear or circular transcripts. Excluding those reads would result in low numbers of reads for normalization, leading to very noisy estimates of enrichment, in particular for HSV-1 infection where linear transcripts are expected to be lost. Including those reads, however, would lead to under-estimation of circRNA enrichment. Since only a relatively small fraction of genes harbour circRNAs, use of junction reads from all genes circumvents this issue. Moreover, there are circRNAs without corresponding linear transcripts, such as CDR1as, one of the few circRNAs with a well-described function [16]. This circRNA would otherwise have to be excluded from our analysis.

Notably, the fraction of linearly spliced host reads decreased considerably during WT infection (S1D Fig). This is largely a consequence of HSV-1-induced read-through transcription. Read-through transcripts are rarely spliced downstream of the poly(A) site and are retained in the nucleus [21, 22], thus escaping *vhs*-mediated decay. In contrast, in *vhs* infection, the fraction of spliced reads did not decrease (S1E Fig) due to reduced read-through transcription and absence of *vhs*-mediated decay of cytosolic mRNAs. Thus, normalization by linear splice junction read counts better quantifies the impact of *vhs*-mediated decay on linear transcripts than normalization by the total number of reads mapped to the host genome as in WT infection a large fraction of host genome reads originate from read-through transcripts that escape *vhs*-mediated decay. Normalized circRNA counts increased substantially with increasing duration of WT infection, indicating increasing enrichment of circRNAs relative to linear transcripts (Fig 2C, S2A–S2C Fig, red line = linear regression estimate over all circRNAs). By 8 h p.i. WT infection, circRNAs were enriched ~6.5-fold in total RNA compared to uninfected cells. In contrast, Δ*vhs* infection exhibited only a very modest ~1.3-fold enrichment in circRNAs by 8 h p.i. (Fig 2D, S2D–S2F Fig). This can likely be attributed to the global loss of host transcriptional activity in HSV-1 infection [51], which affects circRNAs less than linear transcripts as circRNAs are much more stable. Notably, if we instead normalize by the total numbers of reads mapped to the host (as done in the study by Shi *et al.* [19]), we also observe enrichment of circRNAs in WT infection, albeit to a lower degree (S3 Fig).

As noted above, one factor that might negatively impact circRNA detection in the Δ*vhs* infection time-course is the shorter read length. CIRI2 and circRNA_finder require that at least one read overlaps by at least 19 and 20 nt, respectively, on both sides of identified circular junctions. Thus, longer reads could potentially lead to identification of more circRNAs. To address this issue, we pursued two alternative approaches. First, we trimmed reads from the WT infection time-course to 76 nt and applied our *de novo* circRNA detection method to these trimmed reads. Second, we implemented an alternative circRNA detection method based on aligning reads against circRNA junction sequences constructed from annotated circRNAs, which allowed to reduce or increase the required overlap on either side of the junction (see methods and S4 Fig). In this case, false positive rates are reduced by the restriction to annotated circRNAs. Both with trimmed reads and the alignment-based pipeline with a required overlap of only 10 nt, we observed approximately the same circRNA enrichment in WT infection as with the *de novo* detection pipeline on full reads (S5 and S6 Figs). Moreover, since circRNA enrichment is consistent between highly and lowly expressed circRNAs as shown by the low deviation from the linear regression line, exclusion of lowly expressed circRNAs by adapting circRNA detection thresholds to sequencing depth would not alter conclusions. Unless explicitly noted otherwise, all results reported in the following use the alignment-based pipeline requiring an overlap of at least 10 nt on either side of the junction.

We also analyzed data from a separate RNA-seq experiment in which we obtained total RNA for mock, WT, Δ*vhs* and ΔICP27 infection in the same experiment as well as subcellular fractions (chromatin-associated, nucleoplasmic and cytosolic RNA) (all 8 h p.i., n = 2) [18, 22, 52]. This showed enrichment of circRNAs in total RNA of WT and ΔICP27 infection, but not Δ*vhs* infection (S7 Fig), Notably, although linear regression analysis over all circRNAs (red line) estimated only a 1.5- to 2-fold enrichment in WT infection, some of the most highly expressed circRNAs were more strongly enriched. To confirm that enrichment of circRNAs in HSV-1 infection was not due to increased abundance of circRNAs, we also analyzed recently published total RNA-seq data for mock and HSV-1 strain KOS (WT KOS) infection (n = 1 replicate), where ERCC spike-ins were added to allow absolute quantification of expression levels [53]. This confirmed the strong enrichment of circRNAs relative to linear RNAs in a second HSV-1 strain (~10-fold, S8A Fig). However, normalization to ERCC spike-in read counts showed that absolute levels of most circRNAs in total RNA remained largely unchanged during HSV-1 infection (S8B and S8C Fig). We conclude that *vhs*-mediated degradation of linear mRNAs that spares circRNAs leads to an enrichment of circRNAs relative to linear mRNAs during HSV-1 infection.

As circRNA reads that do not cross circular junctions can be aligned by standard RNA-seq mapping programs, they are counted towards circRNA "parent" genes and exons in standard differential gene expression (DGE) and differential exon usage (DEU) analyses. Consequently, genes encoding for circRNAs tend to have significantly higher fold-changes in DGE analysis of total RNA in WT infection from 6 h p.i. (one-sided Wilcoxon rank sum test, p-value <0.0001, S9A Fig). This was not observed in Δ*vhs* infection. Similarly, DEU analysis with DEXSeq showed differential exon usage in WT infection for many exons contained in circRNAs (multiple testing adjusted p-value ≤0.005). By 8 h p.i. WT infection, ~17% of circRNAs showed differential exon usage for at least one exon within the genomic range of the circRNA (Fig 2E), with increased exon usage for almost all of these circRNAs (Fig 2F). This effect was most pronounced for highly expressed circRNAs (S9B Fig). Consistent with the small degree of circRNA enrichment late in Δ*vhs* infection, 3.6% of circRNAs also showed differential exon usage by 8 h p.i. Δ*vhs* infection (S9C and S9D Fig). In summary, our results demonstrate that enrichment of circRNAs in WT infection biases not only differential circRNA analyses, but

also DGE and DEU analyses of total RNA and thus needs to be taken into account for the latter.

## Induction of NEAT1 linear and circular splicing in HSV-1 infection

In contrast to other circRNAs, one circRNA (circBase ID: hsa_circ_0003812) was actually induced in HSV-1 infection even in absence of *vhs*. hsa_circ_0003812 is encoded in the unique part of the long NEAT1_2 transcript (Fig 1C), was essentially absent in mock-infected cells (0–2 reads in 4sU-RNA, total RNA and all subcellular fractions, ≤0.22 normalized circRNA count) and highly abundant in HSV-1 infection (Fig 3A, S10 Fig). ERCC spike-in normalization in mock and WT KOS infection confirmed that abundance of this NEAT1_2 circRNA indeed increased in absolute levels (S8B and S8C Fig), indicating high levels of *de novo* synthesis of this circRNA in HSV-1 infection. Consistent with this, induction of the NEAT1_2 circRNA during HSV-1 infection was also confirmed in samples of very recently transcribed RNA, i.e., 4sU-, nucleoplasmic and chromatin-associated RNA. This is particularly noteworthy considering the small number of circRNAs we recovered in these samples (S1B, S1C, S11 Figs). From 6 h p.i., the NEAT1_2 circRNA was among the most highly expressed circRNAs in total, 4sU-, chromatin-associated and nucleoplasmic RNA in WT, ΔICP27 and Δ*vhs* infection (S10 Fig). Consistent with the nuclear localization of NEAT1, only few circular NEAT1_2 reads were found in cytosolic RNA. Other NEAT1_2 circRNAs were also detected, but only with much fewer reads (Fig 3A). In summary, our data indicate that biogenesis of the hsa_-circ_0003812 NEAT1_2 circRNA is induced during HSV-1 infection.

The hsa_circ_0003812 NEAT1_2 circRNA was previously found to be enriched in RNA-seq data of human fibroblasts after treatment with the 3'->5' RNA exonuclease RNase R [54], confirming that it is a circRNA. Notably, however, NEAT1_2, is partially resistant to RNase R due to the stabilizing triple helical structure at its 3'end, similar to MALAT1 [20]. Thus, the linear NEAT1_2 transcript is not strongly depleted by RNase R treatment even with an improved protocol using A-tailing prior to RNase R treatment and a Li+ buffer (S12A Fig) [20]. Nevertheless, we identified a study of circRNAs in Akata cells, in which the linear NEAT1_2 transcript was fully depleted by RNase R treatment in total and cytoplasmic RNA, yet partially resistant in nucleoplasmic RNA, while the hsa_circ_0003812 circRNA was always retained (S12B Fig) [55]. Interestingly, we observed significant enrichment of the circRNA region in our total RNA-seq time-course of WT infection from 6 h p.i. compared to both up- and downstream regions (Fig 3B, S13A Fig, 1.5- to 1.76-fold increase in the DEXSeq analysis, multiple testing adjusted p<0.005). This effect is likely not mediated by *vhs* since NEAT1_2 is located in the nucleus while *vhs* is active in the cytosol. Furthermore, some enrichment of the circRNA region was also observed in Δ*vhs* infection by 8 h p.i., consistent with an attenuated progression of Δ*vhs* infection (S13B Fig). Both NEAT1_1 and NEAT1_2 were found to be highly unstable in mouse, with half-lives of ~30 and ~60 min, respectively [56]. Thus, enrichment of the NEAT1_2 circRNA in HSV-1 infection is likely due to its high stability compared to the linear transcript combined with high *de novo* synthesis of this circRNA.

The following evidence further confirms that hsa_circ_0003812 is a circRNA. First, it was identified by three distinct algorithms for circRNA detection that employ different approaches: CIRI2, circRNA_finder and our aligment-based approach. Combination of two or more different circRNA detection methods has been reported as highly successful in removing false positive predictions [43]. Second, since false positives in circRNA detection largely originate from repetitive sequences, we performed a BLAST search in the human and HSV-1 genomes for the 20 nt on either side of the circular junction. This confirmed that both sequences are unique. Third, we ran our alignment-based circRNA detection approach with a required overlap of at

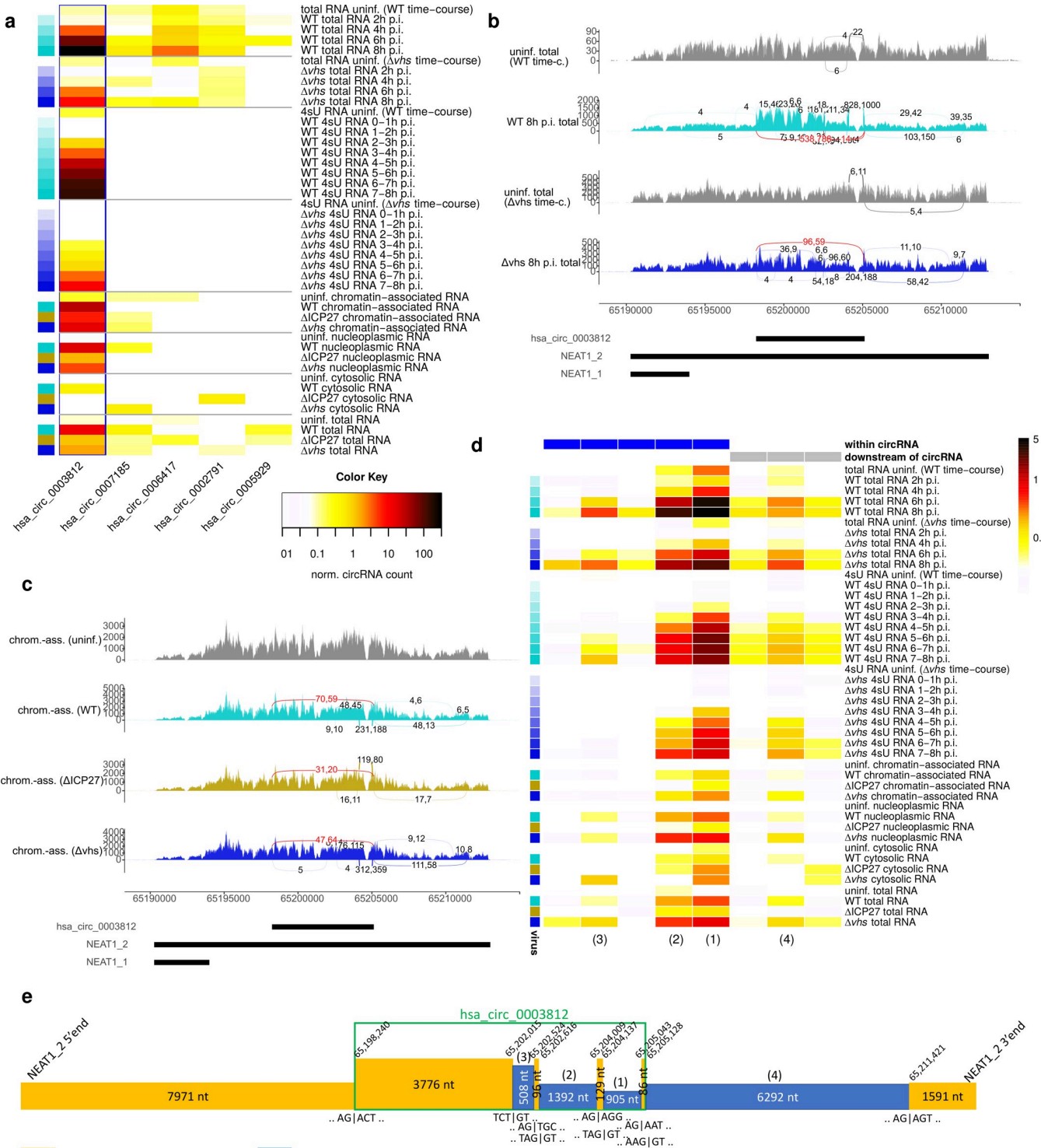

**Fig 3. Induction of a NEAT1_2 circular and linear splicing in WT, Δ*vhs* and ΔICP27 infection. (a)** Heatmap of normalized circRNA counts (normalized to the number of linear junction reads mapped to the host genome) for all identified NEAT1_2 circRNAs in total and 4sU-RNA time-courses of WT and Δ*vhs* infection and nucleoplasmic, chromatin-associated, cytoplasmic and total RNA for 8 h p.i. WT, Δ*vhs* and ΔICP27 infection. Columns represent individual circRNAs, which are ordered according to their genomic coordinates from the most 5' to the most 3'. The hsa_circ_0003812 NEAT1_2 circRNA is marked by a blue rectangle. **(b, c)** Sashimi plots showing NEAT1 read coverage (overlay of both replicates) and circular (red) and linear (same color as read coverage) splice junctions as arcs connecting acceptor and donor splice site in **(b)** selected samples of the total RNA time-courses of WT and Δ*vhs* infection and **(c)** chromatin-associated RNA in mock, WT, Δ*vhs* and ΔICP27 infection. Number of junction reads are annotated to arcs separately for the two replicates. Junctions are only

shown if at least 4 reads align by at least 10 nt on both sides of the junction. Genomic coordinates of NEAT1 transcripts and the HSV-1-induced circRNA are shown at the bottom. Sashimi plots of the full total RNA and 4sU-RNA time-courses and nucleoplasmic and total RNA from the subcellular fractions experiment are shown in S13 and S15 Figs. **(d)** Heatmap of linear splicing rates (= number of reads for a linear junction / number of exon-intron reads crossing the corresponding acceptor and donor splice sites) for the WT and Δ*vhs* infection time-courses and the subcellular fraction experiment (dark red = high, white = low, see color bar on the right). Columns represent splice junctions, which are ordered according to their genome positions, first by the donor and then by the acceptor splice site. Counts include only junction and exon-intron reads overlapping the junction or exon-intron boundary, respectively, by at least 10 nt on either side. A pseudocount of 0.01 was used to avoid division by zero. Rectangles on top indicate whether the splice junction is within the genomic region or downstream of the hsa_circ_0003812 circRNA. Colors on the left indicate the virus and time-point of infection (light colors early in infection, darkest colors 8 h p.i, cyan = WT, blue = Δ*vhs*, gold = ΔICP27). Numbers in brackets at the bottom refer to the four most frequent splice junctions outlined in **(e)**. **(e)** Schematic overview on the circular and the four most frequent linear NEAT1_2 splice junctions induced in HSV-1 infection and sequences around splice sites ("|" indicates exon-intron boundaries in sequences). The height of boxes indicates the relative abundance of each exon (= never or rarely spliced out) and intron (= frequently spliced out) by 8 h p.i. WT infection. Genomic coordinates on chromosome 11 are indicated for the start and end of exons. Numbers in brackets on top of introns indicate the ranking of corresponding splice junctions according to their splicing rate shown in **(d)**. This ranking is also indicated in **(d)** at the bottom.

least 40 nt on either side of circular junctions on the WT 4sU-seq and total RNA-seq time-courses. This still identified up to 170 reads for the NEAT1_2 circRNA in total RNA and up to 44 in 4sU-RNA (17–36% of reads identified with a ≥10 nt overlap) and confirmed the strong induction of this circRNA (S14A and S14B Fig). Finally, we implemented a method to determine so-called "confirming read pairs", i.e., pairs of reads where both reads could be aligned linearly to the genome within a circRNA region but only in crosswise direction that is only consistent with a circRNA but not a linear transcript (S14C Fig). We selected those confirming read pairs for which their genomic distance would imply a fragment size ≥500 nt if they originated from a linear transcript and the fragment size would be smaller if they originated from the circRNA. In this way, we identified up to 45 confirming read pairs per replicate in WT infection for hsa_circ_0003812 (S14B Fig).

NEAT1_2 circular splicing was accompanied by linear splicing both within and downstream of the circRNA region but not within the NEAT1_1 region (Fig 3B and 3C, S13 and S15 Figs). None of these linear splicing events were annotated, but some were already observed at very low rates in uninfected cells. As read counts for the most frequent linear splice sites were comparable or even higher than for the induced hsa_circ_0003812 circular junction, NEAT1_2 linear splicing was also increased in absolute levels. Calculation of splicing rates (= number of junction reads / number of exon-intron reads crossing the corresponding acceptor and donor splice sites) showed that linear splicing was indeed induced and not simply observed more frequently due to the previously reported up-regulation of NEAT1 transcription in HSV-1 infection [29, 31] (Fig 3D). As linear splicing was also induced downstream of identified circRNAs and several of the confirming read pairs for the hsa_circ_0003812 circRNA included linear splice junctions, both linear and circular NEAT1_2 transcripts were further spliced in HSV-1 infection. Canonical GT-AG splice signals were used both for the circular and the most frequent linear splice junctions (see Fig 3E for a diagram) and they tended to be conserved at least among primates (S16 Fig). Since circRNA biogenesis by back-splicing requires the spliceosome [11], our data suggest that circular and linear splicing of NEAT1_2 are linked, with the circRNA originating from back-splicing during NEAT1_2 linear splicing (see also Fig 1A). Splicing rates and their increases compared to mock differed substantially between the different linear splice junctions within the circRNA region (Fig 3D), indicating presence of alternative splicing isoforms of the circRNA. Induction of linear splicing was also observed in 4sU-, chromatin-associated and nucleoplasmic RNA and in ΔICP27 and Δ*vhs* infection (Fig 3C and 3D, S15 Fig), although it was less pronounced in ΔICP27 infection. In summary, both linear and circular splicing of NEAT1_2 is induced during HSV-1 infection while the RNA is still associated with the chromatin. Although ICP27 is not required for NEAT1_2 splicing, it may contribute.

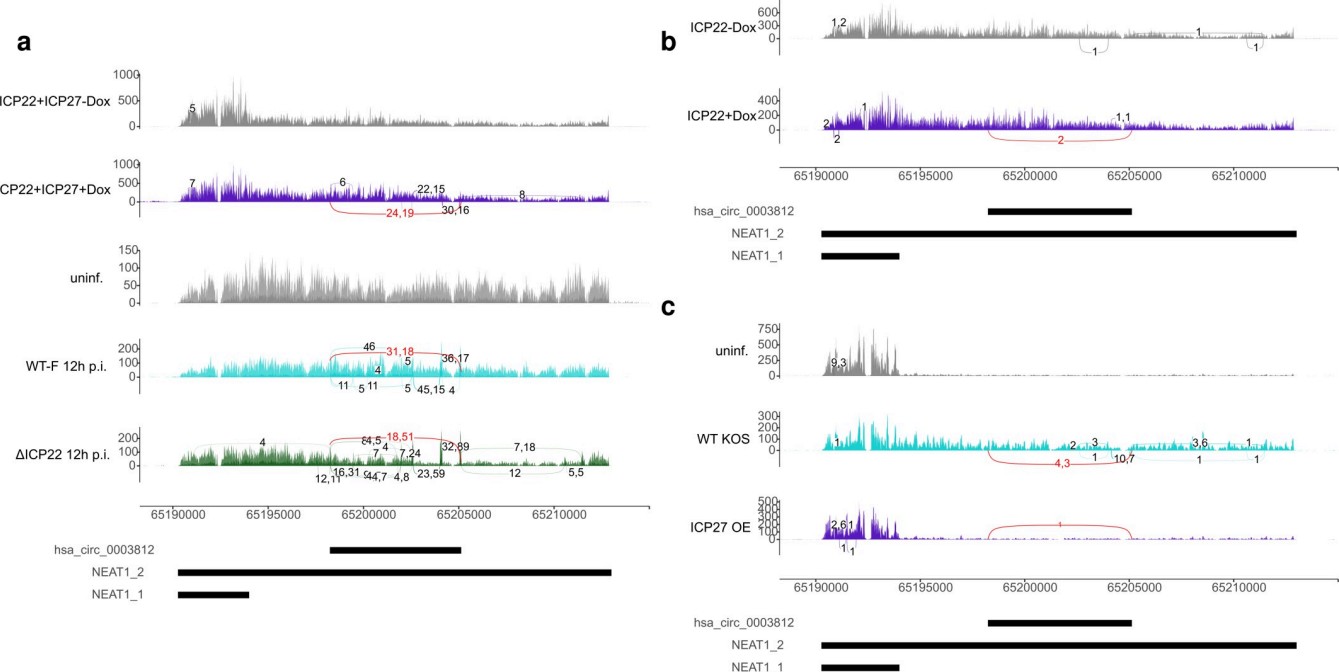

**Fig 4. Ectopic co-expression of ICP22 and ICP27 is sufficient for induction of NEAT1_2 splicing.** Sashimi plots as in Fig 3 showing read coverage and circular and linear splice junctions in **(a)** T-HFs-ICP22/ICP27 cells without and with Dox-induced co-expression of ICP22 and ICP27 as well as mock, WT-F and ΔICP22 infection at 12 h p.i., **(b)** T-HFs-ICP22 cells without and with Dox-induced ICP22 expression and **(c)** mock and WT KOS infection and ICP27 overexpression. A minimum read count of 4 was again required for **(a)**, while in **(b)** and **(c)** all junction reads are shown covering at least 10nt on either side of the junction. For an explanation of sashimi plots, see caption to Fig 3. Circular splice junctions are marked in red, linear splice junctions in the same color as read coverage.

Recently, we also performed total RNA-seq of infection with mock, HSV-1 strain F (WT-F) and its ΔICP22 mutant at 8 and 12 h p.i. (n = 2) [57]. We furthermore generated telomerase-immortalized human foreskin fibroblasts (T-HFs) that express either ICP22 in isolation (T-HFs-ICP22 cells) or in combination with ICP27 (T-HFs-ICP22/ICP27 cells) upon doxycyclin (Dox) exposure. Total RNA-seq of T-HFs-ICP22 and T-HFs-ICP22/ICP27 cells was performed both with and without Dox exposure (n = 2). Ectopic co-expression of ICP22 and ICP27 lead to induction of both circular and linear splicing of NEAT1_2 (Fig 4A). However, ICP22 knockout did not abolish HSV-1-induced NEAT1_2 circular and linear splicing (Fig 4A, S17 Fig), thus ICP22 expression is not required. This also confirmed induction of linear and circular NEAT1_2 splicing in a third HSV-1 strain. Although circRNA reads were recovered after Dox-induced expression of ICP22, this was limited to 2 reads in one replicate (Fig 4B).

To investigate whether ICP27 expression alone could induce NEAT1_2 splicing, we analyzed our recently published 4sU-seq data of mock and WT KOS infection and of cells transfected with an ICP27-expressing plasmid [34]. This again confirmed induction of both linear and circular NEAT1_2 splicing in WT KOS infection but recovered only one circular read upon ICP27 overexpression in one replicate (Fig 4C). Notably, NEAT1 was upregulated upon ectopic co-expression of ICP22 and ICP27 (2.2-fold, $p < 10^{-11}$) and in ΔICP22 (2.1-fold, $p < 10^{-4}$) and ΔICP27 (4.9-fold, $p < 10^{-12}$) infection, but not upon expression of either ICP22 or ICP27 alone. In summary, our results show that neither ICP27 nor ICP22 are required for induction of linear and circular NEAT1_2 splicing but co-expression of both proteins is sufficient. Considering the small number of circular reads recovered when expressing either one of

these proteins alone, we can neither confidently confirm nor exclude that either ICP22 or ICP27 alone may be sufficient to (weakly) induce NEAT1_2 circular splicing.

## Influenza A virus infection also induces NEAT1_2 splicing

Since NEAT1_2 is also up-regulated in IAV infection [29], we investigated presence of circRNAs in previously published total RNA-seq time-courses of influenza A/California/04/09 (H1N1), A/Wyoming/03/03 (H3N2), and A/Vietnam/1203/04 (H5N1) HALo virus infection (3, 6, 12 and 18 h p.i. plus time-matched controls) of human tracheobronchial epithelial cells (HTBE, multiplicity of infection (MOI) = 5) and monocyte-derived macrophages (MDM, MOI = 2) [41]. The H5N1 HALo mutant virus is an attenuated virus generated from WT influenza A/Vietnam/1203/04. H1N1 infection of HTBE cells included 24 h p.i. and two H5N1 infection time-courses of MDM cells were performed. This revealed a strong enrichment of NEAT1_2 circular splicing in H5N1 infection of both cell types and H3N2 infection of MDM cells as well as a weak enrichment in H3N2 infection of HTBE cells and H1N1 infection of MDM cells (Fig 5A, S18 Fig). The most highly expressed NEAT1_2 circRNA was again hsa_-circ_0003812 circRNA as for HSV-1 infection (marked by a blue rectangle in Fig 5A) and its

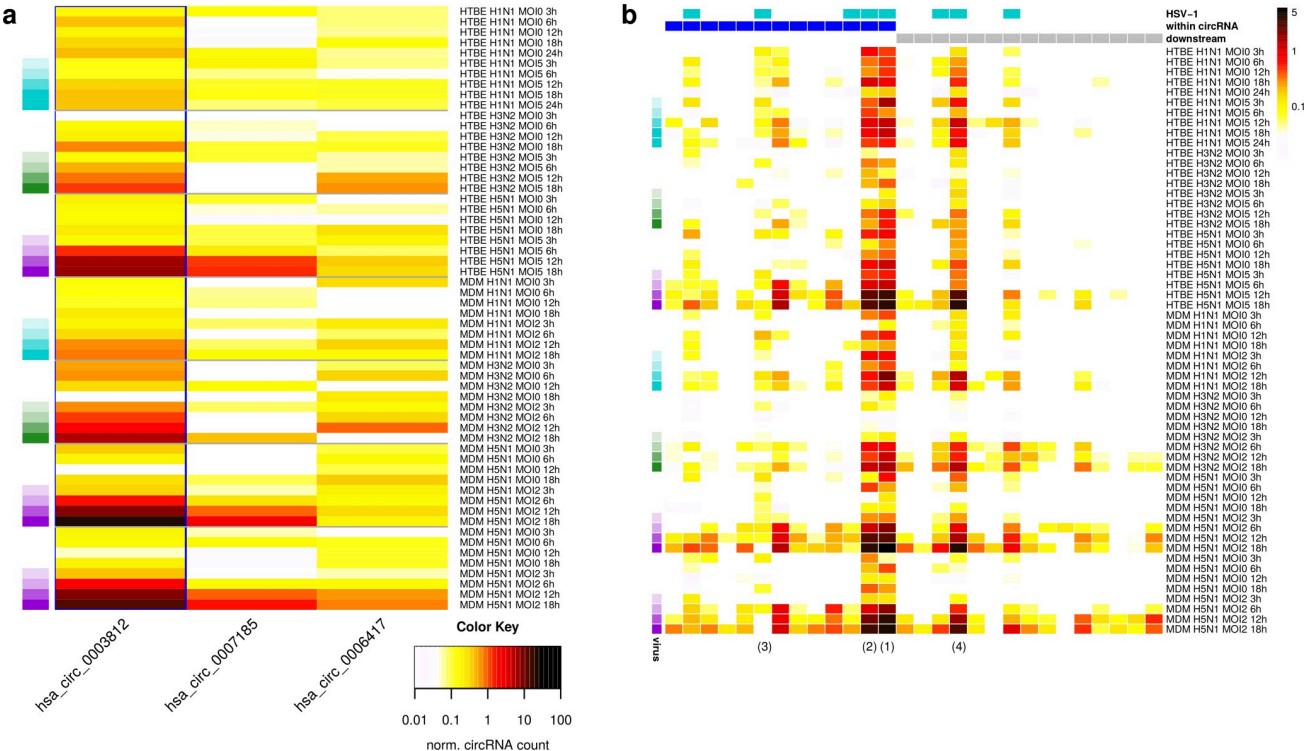

**Fig 5. Induction of NEAT1_2 circular and linear splicing in IAV infection. (a)** Heatmap of normalized circRNA counts (normalized to number of linear splice junctions mapped to the host genome) for the three most highly expressed NEAT1_2 circRNAs identified in H1N1, H3N2 and H5N1 infection of HTBE and MDM cells or time-matched controls. Columns represent individual circRNAs, which are ordered according to their genomic coordinates from the most 5' to the most 3'. The NEAT1_2 hsa_circ_0003812 also induced in HSV-1 infection is marked by a blue rectangle. IAV strain and time-points of infection are color-coded on the left (white = time-matched control, cyan = H1N1, green = H3N2, purple = H5N1, darker colors indicate later time-points). **(b)** Heatmap of linear splicing rates (= number of reads for a linear junction / number of exon-intron reads crossing the corresponding acceptor and donor splice sites) in IAV infection and time-matched controls (dark red = high, white = low, see color bar on the right). Columns represent individual splice junctions, which are ordered according to their genome positions. Counts include only junction and exon-intron reads overlapping the junction or exon-intron boundary, respectively, by at least 10 nt on either side. A pseudocount of 0.01 was used to avoid division by zero. Colors on the left indicate time-points of infection as in **(a)**. Colored rectangles on top indicate whether linear splice junctions were also observed in HSV-1 infection (cyan) or are within (blue) or downstream (gray) of the hsa_circ_0003812 circRNA region. Numbers in brackets at the bottom refer to the introns outlined in the schematic in Fig 3E.

enrichment increased with the duration of infection. Similar trends were observed for linear splicing events, both within and downstream of the circRNA region (Fig 5B). Notably, both the hsa_circ_0003812 circRNA and linear splicing events were also detected in some of the time-matched controls, however normalized circRNA counts and linear splicing rates were significantly lower than in infected cells (with the exception of H1N1 infection, S18A Fig).

IAV infection also induces the degradation of host RNAs via two mechanisms. Cap-snatching by the viral RNA-dependent RNA polymerase leads to cleavage of host RNAs downstream of the cap and is required for synthesis of viral mRNAs [58]. The PA-X protein produced by all IAV strains [59] selectively degrades RNAs transcribed by Pol II, including long non-coding RNAs, and is localized and acts predominantly in the nucleus [60]. To exclude that enrichment of the NEAT1_2 circRNA is simply due to degradation of linear transcripts without increased biogenesis of the circRNA, we also investigated overall enrichment of circRNAs in IAV infections using linear regression analysis as for HSV-1 infection. This indeed showed an enrichment of circRNAs by up to 1.4-fold in H1N1 infection, 6.8-fold in H3N2 infection and 4.6-fold for H5N1 infection (S19A–S19G Fig). It also revealed a much higher enrichment for several other circRNAs, such as a circRNA of the ZC3HAV1 (zinc finger CCCH-type containing, antiviral 1) gene. ZC3HAV1 is induced during influenza A/WSN/33 (WSN/H1N1) infection and ectopic expression of ZC3HAV1 inhibits WSN replication [61]. Thus, the ZC3HAV1 circRNA is likely increasingly produced during IAV infection due to up-regulation of ZC3HAV1 transcription and associated increased circRNA biosynthesis and is not only enriched due to linear host RNA degradation. We compared fold-changes in normalized circRNA counts between infection and time-matched controls for 7077 well-expressed circRNAs (normalized circRNA count >1 in at least one condition), which showed that the hsa_circ_0003812 NEAT1_2 circRNA was among the 1% most enriched circRNAs in H3N2 and H5N1 infection of MDM cells (S19H Fig). Occasionally, it was even more enriched than the ZC3HAV1 circRNA. Moreover, splicing rates of linear NEAT1_2 transcripts, which should not escape host mRNA degradation in IAV infection, also increased during infection (Fig 5B). Thus, our results indicate that–at least in H3N2 and H5N1 infection–enrichment of NEAT1_2 circular splicing results from increased biogenesis of the circRNA and not only from IAV-induced degradation of host mRNAs.

Since both HSV-1 and IAV infection disrupt transcription termination, we also evaluated read-through for IAV infection to correlate this to the extent of induction of NEAT1_2 splicing. Paralleling NEAT1_2 induction, read-through was higher in MDM cells than in HTBE cells and higher in H5N1 infection than in H3N2 infection (S20 Fig), while H1N1 infection led to the lowest extent of read-through. Disruption of transcription termination in IAV infection is mediated by the viral NS1 protein via inhibition of the CPSF30 subunit of the cleavage and polyadenylation specificity factor (CPSF) [62]. Our results are consistent with previous reports that the NS1 protein of A/California/04/09 H1N1 virus cannot bind and inhibit CPSF30 [63], while NS1 of the H3N2 and H5N1 strains can [64, 65]. However, a recent report indicated that the NS1-CPSF30 interaction is not necessary for disruption of transcription termination in IAV infection [40]. This could explain why read-through is still observed in H1N1 infection. A similar observation can be made in HSV-1 infection. Although HSV-1-induced disruption of transcription termination is mediated by ICP27 via interaction with CPSF subunits [34], reduced but significant read-through is also observed in ΔICP27 infection likely as a stress response [22, 39]. Due to this correlation between induction of NEAT1_2 splicing and read-through and a recent report that heat stress also up-regulates NEAT1_2 expression [38], we also analyzed our previously published 4sU-seq data of 1 and 2 h salt and heat stress regarding NEAT1_2 splicing. However, neither circular nor linear NEAT1_2 splicing was induced in either of the two stress conditions.

## CDK7 inhibition induces NEAT1_2 splicing and disrupts transcription termination in cancer cell lines

To shed additional light on the molecular mechanism(s) governing NEAT1_2 splicing, we searched the recount3 database [66] using Snaptron [67] for presence of the novel linear NEAT1_2 splicing events. Recount3 provides read coverage data for >700,000 publicly available human and mouse RNA-seq samples from the Sequence Read Archive (SRA). Snaptron allows rapidly searching samples in recount3 for (linear) splice junctions within specific genomic regions. Our Snaptron query recovered 6,391 human samples in which at least one of the four most frequent NEAT1_2 linear splice junctions (see Fig 3E) induced in HSV-1 infection was covered by ≥10 reads. This included our HSV-1 RNA-seq experiments, several other studies on HSV-1 infection, the IAV infection time-courses, 62 studies focusing on circRNAs and 21 studies using RNase R (S21A Fig). The latter provide further support for the association of linear and circular splicing of NEAT1_2. Particularly high numbers of NEAT1_2 splice junctions were found in samples of blood cells, such as platelets, erythrocytes, peripheral blood mononuclear cells (PBMCs), erythroleukemia (K562) cells, and more (S21A Fig). Consistently, one of these studies reported that circRNAs are strongly enriched in platelets and erythrocytes [68], which both lack a cell nucleus and thus *de novo* transcription. In absence of *de novo* transcription, the high stability of circRNAs leads to their enrichment relative to linear RNAs.

Application of our circRNA alignment pipeline to two RNA-seq experiments for platelets [69] and erythrocytes [70] identified multiple NEAT1_2 circRNAs including the hsa_-circ_0003812 circRNA induced by HSV-1 infection (S21B Fig). Interestingly, almost no expression was detected in the ~4 kb between the 3'end of the NEAT1_1 transcript and the 5'end of hsa_circ_0003812, however, expression extended beyond hsa_circ_0003812. This either implied novel linear transcripts beginning near the 5'end of hsa_circ_0003812 or a novel circRNA not included in circBase. Indeed, our *de novo* circRNA detection pipeline identified a novel circular junction connecting the 5' end of hsa_circ_0003812 with the 3'end of the downstream expressed region. This confirms that only circular, but not linear, NEAT1_2 transcripts were present in these cells and that these circRNAs are also linearly spliced. It also shows that NEAT1_2 circRNAs naturally occur in uninfected cells, however likely at such low levels that they only become detectable upon degradation of linear transcripts. It should be repeated that NEAT1_2 has been shown to be up-regulated in HSV-1 and IAV infection by RT-qPCR [29], thus enrichment of NEAT1_2 circular splicing in HSV-1 and IAV infection cannot be explained by loss of *de novo* transcription followed by degradation of linear transcripts. However, this partly explains the presence of NEAT1_2 circular and linear junctions in uninfected cells from the IAV infection time-courses as MDM cells were obtained from blood.

Interestingly, the Snaptron search also identified NEAT1_2 linear splicing after inhibition of CDK7 by THZ1 in multiple cancer cell lines, which was not observed in untreated cells. CDK7 is an essential part of the TFIIH transcription factor complex, phosphorylates Pol II CTD at Ser5 residues and plays key roles in maintenance of promoter-proximal Poll II pausing and the regulation of transcription elongation [71]. Overexpression of CDK7 has been associated with a number of cancers and correlated to poor prognosis [71]. THZ1 is a highly selective covalent inhibitor of CDK7 with anti-tumor activity against multiple different types of cancer [72], but also inhibits CDK12 and CDK13 at higher concentrations [73]. Experiments identified with the Snaptron search cover prostate cancer (VCaP, LNCaP and DU145 cells) [42], bladder cancer (HCV-29 cells) [74], esophageal squamous cell carcinoma (TE7 and KYSE510 cells) [75], nasopharyngeal carcinoma (C666-1, HK1 and HNE1 cells) [76], pancreatic ductal adenocarcinoma (BxPC3, MiaPaCa-2 and PANC1 cells) [77] and chordoma (UM-Chor1 cells) [78]. Further literature search identified a second study of chordoma cells (UM-Chor1 and

CH22) [79] with up to 24 h THZ1 treatment and a study on THZ1 treatment of B-cell acute lymphocytic leukemia cells (Nalm6) [80]. Realignment of corresponding samples confirmed linear splicing upon THZ1 treatment in all cell lines apart from DU145 and showed that splicing rates increased with the duration and dosage of THZ1 treatment (Fig 6A and 6B and S22A Fig).

Surprisingly, considering the previously observed strong association between linear and circular NEAT1_2 splicing, no NEAT1_2 circRNAs were recovered for any of these samples neither with our alignment-based nor our *de novo* circRNA discovery pipeline. However, in all cases almost no circRNAs (maximum 105 circRNAs, mean 18) were recovered at all, not even the most ubiquitous circRNAs such as HIPK3, CORO1C or ASXL1. As read length was ≥75 nt for all samples, this cannot be explained by short read length. It rather suggests that circRNAs were generally not or only poorly recovered in all these studies. While a literature search identified no other studies for specific CDK7 inhibition without poly(A) selection, the Snaptron search also recovered linear NEAT1_2 splicing in chromatin-associated RNA of HEK293 cells upon 2 h treatment with DRB [81]. DRB inhibits CDK7 [82] but also the CDK9 kinase component of the positive P-TEFb complex [83] and other kinases [84]. The hsa_-circ_0003812 NEAT1_2 circRNA induced by HSV-1 and IAV infection was indeed observed upon DRB treatment, but not any of the other NEAT1_2 circRNAs present in platelets and erythrocytes (S22B Fig). This provides some evidence that CDK7 inhibition leads to NEAT1_2 circular splicing.

In any case, identification of linear splicing in absence of circRNAs shows that the linear NEAT1_2 isoform is spliced upon CDK7 inhibition. As no circRNAs were recovered, induction of linear splicing cannot be an artefact of circRNAs being enriched after THZ1-mediated transcription inhibition. Furthermore, spliced linear NEAT1_2 transcripts should not be more or less stable than unspliced NEAT1_2 transcripts as splicing occurs upstream of the last 100 nt of NEAT1_2 that form the stabilizing helical structure. Interestingly, however, CDK7 inhibition led to a relative increase of NEAT1_2 compared to NEAT1_1, suggesting that NEAT1_2 transcription was less strongly reduced by CDK7 inhibition than NEAT1_2 transcription or potentially even increased. To exclude that inhibition of transcription *per se* leads to NEAT1_2 splicing, we investigated data from one of the UM-Chor1 experiments that included actinomycin D (Act-D) treatment, which inhibits transcription but not via inhibition of cyclin-dependent kinases (CDKs) [85]. Act-D treatment, while leading to a similar relative increase in the abundance of NEAT1_2 compared to NEAT1_1 as THZ1 treatment, did not induce NEAT1_2 splicing (S22A Fig). In addition, siRNA-mediated knockdown of CDK7 in VCaP and LNCaP cells also led to NEAT1_2 splicing and a relative increase in NEAT1_2 compared to NEAT1_1 (Fig 6C), including many more novel splicing events. In contrast to splicing observed in HSV-1 and IAV infection and upon THZ1 treatment, some of these splicing events extended into the NEAT1_1 region. This indicates that indeed inhibition of CDK7, and not of other CDKs, leads to NEAT1_2 splicing. Interestingly, siRNA-mediated knockdown of the MED1 component of the Mediator complex in VCaP and LNCaP cells had the same effect (Fig 6C). CDK7 directly phosphorylates MED1 at Threonine 1457 [42], which together with a phosphorylation at Threonine 1032 promotes MED1 association with the Mediator complex [86]. Mediator is responsible for communicating regulatory signals from gene-specific transcription factors to Pol II and in this way impacts transcription at multiple stages [87]. Our results suggest that the effect of CDK7 inhibition on NEAT1_2 splicing may follow from the loss of MED1 phosphorylation and thus loss of MED1 association with Mediator.

Interestingly, CDK7 inhibition also induced read-through transcription, albeit to a lesser degree than HSV-1 or H5N1 infection (Fig 6D). Read-through transcription after THZ1 treatment was comparable to H1N1 or H3N2 infection of HTBE cells, most pronounced in

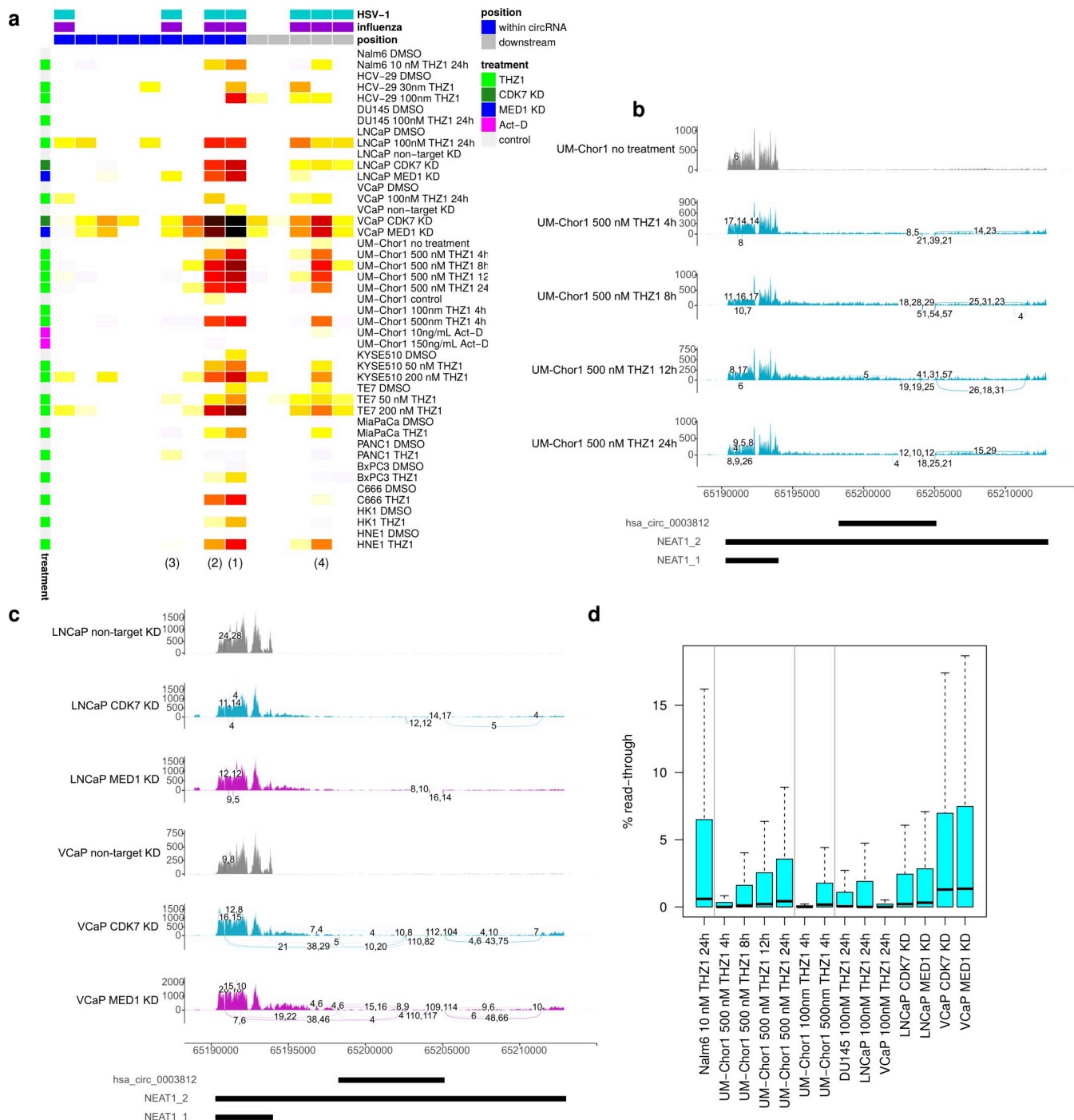

**Fig 6. NEAT1_2 splicing and read-through upon CDK7 inhibition. (a)** Heatmap of linear splicing rates (= number of reads for a linear junction / number of exon-intron reads crossing the corresponding acceptor and donor splice sites) upon treatment by DMSO/control, THZ1 or Act-D or knockdown (KD) of CDK7 or MED1 (dark red = high, white = low, see color bar on the right). Columns represent individual splice junctions, which are ordered according to their genome positions. Counts include only junction and exon-intron reads overlapping the junction or exon-intron boundary, respectively, by at least 10 nt on either side. A pseudocount of 0.01 was used to avoid division by zero. Colored rectangles on top indicate whether linear splice junctions were also observed in HSV-1 (cyan) or IAV (purple) infection or are within (blue) or downstream (gray) of the hsa_circ_0003812 circRNA region. Numbers in brackets at the bottom refer to the introns outlined in the schematic in Fig 3E. **(b, c)** Sashimi plots as in Fig 3 showing read coverage and linear splice junctions in **(b)** UM-Chor1 cells after 4, 8, 12 and 24 h THZ1 or control treatment (3 replicates each) and **(c)** CDK7 and MED1 knockdown in VCaP and LNCaP cells (2 replicates each). All junctions are shown that are covered by at least 4 reads aligning to at least 10 nt on either side of the junction. **(d)** Boxplots showing distribution of %read-through (for calculation see methods) after THZ1 treatment in example studies or CDK7 or MED1 knockdown in VCaP and LNCaP cells.

THZ1-treated Nalm6 cells and increased with the duration and dosage of THZ1 treatment in UM-Chor1 cells. In addition, both CDK7 and MED1 knockdown induced read-through at levels comparable to THZ1-treated Nalm6 cells. Moreover, THZ1 treatment also resulted in extended antisense transcription at promoters. In summary, our results show that CDK7 inhibition impacts RNA transcription and processing at multiple levels.

## Discussion

In this study, we report on novel circular and linear splicing of the long NEAT1_2 isoform, which is induced in both HSV-1 and IAV infection. We focused on NEAT1_2 in this study for three reasons: (i) The NEAT1_2 circRNA was the only circRNA for which levels were clearly and massively increased in HSV-1 infection and which was not simply enriched by *vhs*-mediated degradation of linear RNAs. (ii) It was one of the few circRNAs observed at high levels in newly transcribed 4sU-, nucleoplasmic and chromatin-associated RNA during HSV-1 infection. In contrast to these other few circRNAs, however, the NEAT1_2 circRNA is not or rarely found in uninfected cells. This indicates high levels of *de novo* synthesis of this circRNA during HSV-1 infection. (iii) NEAT1_2 has not previously been reported to be spliced either in a circular or linear fashion except for one linear splicing event towards the 3'end of NEAT1_2. This splicing event represented a shorter 793 nt version of the last "intron" in Fig 3E with the same 3' but a different 5' splice site and was recently reported to increase extractability of NEAT1_2 from the protein phase [88]. Although it was also induced in HSV-1 and IAV infection, it was not among the most frequent splicing events. Re-alignment of the RNA-seq data by Chujo *et al*. confirmed this splicing event as well as splicing of the full last "intron" with very few reads (≤3), but no other splicing.

As NEAT1_2 participates in many RNA-protein interactions in paraspeckles, its unspliced form can only be poorly extracted from the protein phase [88]. Similar observations were made for other architectural RNAs (arcRNAs) of nuclear bodies. As genomic deletion of the 793 nt intron identified by Chujo *et al*. did not significantly affect extractability, splicing itself appeared to be responsible for increased extractability. Chujo *et al*. concluded that NEAT1_2 as well as other arcRNAs must remain unspliced to allow formation of corresponding nuclear bodies. They hypothesized that competition by protein components of paraspeckles for binding of NEAT1_2 normally prevents association of splicing factors with NEAT1_2 and thus NEAT1_2 splicing. This raises the possibility that NEAT1_2 splicing during HSV-1 and IAV infection may simply be a by-product of NEAT1_2 up-regulation and thus increased availability of NEAT1_2 and consequently reduced competition between splicing factors and paraspeckle protein components. Reduced competition would also be consistent with the observation that HSV-1 infection does not alter levels of paraspeckle proteins [29]. Furthermore, NEAT1_2 binds to serine and arginine rich splicing factor 2 (SRSF2) in HSV-1 infection and is required for association of SRSF2 with promoters of the HSV-1 genes ICP0 and thymidine kinase (TK), which are upregulated by SRSF2 [89]. However, the SRSF2 binding sites in NEAT1_2 that are enriched upon HSV-1 infection are upstream of the HSV-1-induced circRNA. Thus, increased binding between SRSF2 and NEAT1_2 is likely not responsible for NEAT1_2 splicing. Moreover, this means that the linear NEAT1_2 transcript is required for SRSF2 association.

We demonstrated that the two immediate-early HSV-1 proteins ICP22 and ICP27 were sufficient to induce NEAT1_2 circular and linear splicing. As outlined in the introduction, both proteins are known to be involved in processes that impact splicing [32, 33, 36]. However, neither of these two proteins were required nor sufficient on their own for (significant) induction of NEAT1_2 splicing. A possible explanation for this observation could be that NEAT1 was

up-regulated upon co-expression of ICP22 and ICP27 and in null-mutant infections of either protein, but not upon expression of either ICP22 or ICP27 alone. Thus, up-regulation of NEAT1_2 could be the trigger for NEAT1_2 splicing rather than specific functions of ICP22 and ICP27. One line of evidence arguing against NEAT1_2 splicing being only a by-product of NEAT1_2 up-regulation is the absence of NEAT1_2 circular or linear splicing in heat stress despite NEAT1_2 also being up-regulated in the heat shock response [38].

Previously, NEAT1_2 knockdown in HeLa cells was shown to reduce HSV-1 glycoprotein density and intensity and inhibit plaque formation, an indicator of mature virus production [31]. The vital role of NEAT1_2 for HSV-1 infection was also confirmed *in vivo* as a thermo-sensitive gel containing NEAT1_2 siRNA could heal HSV-1-induced skin lesions in mice [31]. This siRNA targeted the first "exon" within the circRNA (see Fig 3E), which is contained in both linear and circular spliced and unspliced NEAT1_2 isoforms, thus it should deplete all of them. Interestingly, eCLIP data from ENCODE for the two core paraspeckle proteins SFPQ and NONO shows enriched binding of these proteins in NEAT1_2 within the two most frequently spliced "introns" (1 and 2 in Fig 3E), the "exon" between them, upstream of the circRNA 5' end and towards the 3'end of NEAT1_2, but not in the first two "exons" and "intron" 3. Thus, splicing removes parts of the NEAT1_2 RNA that are important for binding by SFPQ and NONO, which are required for paraspeckle integrity [23]. Moreover, spatial organization of NEAT1_2 in paraspeckles is highly ordered, with 5' and 3' ends of NEAT1_2 confined to the periphery and its central sequences localized in the core of paraspeckles [90]. This makes it unlikely that either spliced linear or circular NEAT1_2 isoforms can provide scaffolds for paraspeckle assembly. Accordingly, up-regulation of NEAT1_2 splicing by unknown mechanisms could also be part of the immune response against HSV-1 to at least dampen paraspeckle up-regulation. On the other hand, SFPQ and NONO have recently been reported to bind around circRNA loci, and SFPQ depletion led to reduced expression of a subset of circRNAs [91]. Thus, these paraspeckle proteins itself could be involved in NEAT1_2 splicing.

Since NEAT1 is up-regulated in both HSV-1 and IAV infection and NEAT1_2 plays a general antiviral role by mediating IL-8 up-regulation, NEAT1_2 splicing observed in both HSV-1 and IAV infection could be mediated by similar mechanisms or play similar roles. However, while NEAT1 is known to have a proviral function in HSV-1 as outlined above, no proviral function of NEAT1 has been reported for IAV infection. Thus, the impact of NEAT1_2 splicing in HSV-1 and IAV infection likely differs at least partly. Linear NEAT1_2 splicing, but likely also circular splicing, is also observed upon CDK7 inhibition in cancer cell lines and NEAT1_2 circRNAs are abundantly found in various blood cells in absence of *de novo* transcription. However, transcription inhibition alone cannot explain NEAT1_2 splicing in HSV-1 and IAV infection and upon CDK7 inhibition as NEAT1 is up-regulated in HSV-1 and IAV infection and transcription inhibition by Act-D does not induce NEAT1_2 splicing. Since CDK7 inhibition leads to a relative increase of NEAT1_2 compared to NEAT1_1, NEAT1_2 expression may not actually be substantially reduced or even increased upon CDK7 inhibition. This again raises the possibility that an increase in NEAT1_2 upon CDK7 inhibition reduces the competition between paraspeckle proteins and splicing factors, resulting in NEAT1_2 splicing.

On the other hand, DRB treatment, which also inhibits CDK7 among other CDKs and leads to NEAT1_2 circular and linear splicing, induces dissociation of paraspeckle proteins from NEAT1_2 and disappearance of paraspeckles [23, 92]. Considering the likely negative impact of NEAT1_2 splicing on paraspeckle formation discussed above, NEAT1_2 splicing could thus contribute to loss of paraspeckles upon DRB treatment. However, paraspeckle disassembly is also observed upon Act-D treatment, which does not induce NEAT1_2 splicing [23]. In contrast, both HSV-1 and IAV infection were found to induce excess formation of

paraspeckles, although in IAV infection they were slightly diffuse [29]. Thus, paraspeckles can be increased even in presence of NEAT1_2 splicing. Considering the substantial induction of NEAT1_2 levels in HSV-1 and IAV infection [29], levels of unspliced NEAT1_2 are likely still sufficiently increased–despite a loss to splicing–to induce excess paraspeckle formation. Alternatively, however, NEAT1_2 splicing may only occur in HSV-1 and IAV infection because NEAT1_2 is more up-regulated than paraspeckle proteins, making NEAT1_2 accessible to splicing factors. Nevertheless, it is tempting to speculate that diffuseness of paraspeckles in IAV infection could at least be partly due to altered paraspeckle formation on (partly) spliced NEAT1_2 isoforms, as NEAT1_2 was not as strongly up-regulated upon IAV infection as upon HSV-1 infection.

Interestingly, we found that selective CDK7 inhibition by THZ1 also disrupts transcription termination in cancer cell lines, albeit less strongly than HSV-1 and H5N1 infection. This was surprising considering a recent report that 1 h THZ1 treatment suppresses Pol II read-through at gene 3'ends in acute myeloid leukemia [93]. It is, however, consistent with earlier reports of impaired transcription termination and 3′-end processing of an snRNA, a polyadenylated mRNA and a histone RNA upon CDK7 inhibition [94, 95]. Our results thus demonstrate for the first time widespread disruption of transcription termination upon CDK7 inhibition. Since read-through transcription was only observed upon long-term or high-dose THZ1 treatment or CDK7 knockdown, a possible explanation for the discrepancy to the study by Sampathi *et al.* is that disruption of transcription termination upon CDK7 inhibition is a downstream response. Interestingly, the relative increase in expression of the non-polyadenylated NEAT1_2 isoform compared to the polyadenylated NEAT1_1 isoform observed upon CDK7 inhibition was also observed in HSV-1 and IAV infection, although not consistently. This raises the possibility that read-through transcription of the NEAT1_1 poly(A) site may be involved in induction of NEAT1_2 splicing by increasing abundance of NEAT1_2. Although salt and heat stress did not induce NEAT1_2 splicing, this does not exclude a role of read-through transcription. While there are strong overlaps between the genes affected by read-through in HSV-1 infection and stress conditions, there are also clear context- and condition-specific differences [22]. Furthermore, the mechanisms underlying disruption of transcription termination in stress conditions remain elusive. This contrasts with HSV-1 and IAV infection, where the HSV-1 ICP27 protein and the IAV NS1 protein have been shown to disrupt transcription termination via interaction with CPSF subunits [34, 41, 62]. It is important to note that our analyses of NEAT1_2 splicing and read-through upon CDK7 inhibition were performed in cancer cell lines, which are particularly susceptible to CDK7 inhibition [72]. Dysregulation of alternative polyadenylation, often associated with wide-spread shortening of 3'UTRs, is common in cancers [96]. NEAT1 also plays different roles in cancer development and both NEAT1 isoforms have been proposed as cancer biomarkers [97], with NEAT1_2 considered a tumor suppressor and NEAT1_1 considered to be oncogenic [98]. Thus, read-through, relative increase of NEAT1_2 expression, or NEAT1_2 splicing upon CDK7 inhibition could be linked to susceptibility of cancer cell lines to CDK7 inhibition.

Our study identified several parallels between HSV-1 and IAV infection, on the one hand, and CDK7 inhibition, on the other hand. This not only includes read-through transcription and NEAT1_2 splicing, but also induction of antisense transcription upon CDK7 inhibition. We previously reported on widespread activation of antisense transcription in HSV-1 infection [99]. Moreover, widespread loss of promoter-proximal pausing was recently reported both upon CDK7 inhibition [93, 100] and HSV-1 infection [101]. Previously, LDC4297A, a different selective inhibitor of CDK7, has been shown to have strong antiviral activity against HSV-1 as well as other herpesviruses (but only low efficacy against H1N1) [102] and CDK7 has been found to be a target of CDK inhibitors inhibiting HSV replication [103]. Both studies

confirm a role of CDK7 in HSV-1 infection. It is tempting to speculate that recruitment of CDK7-containing complexes, similar to recruitment of elongation factors like FACT by ICP22 [104], to the HSV-1 genome may lead to a depletion of CDK7 on host genes, thus mimicking the effects of CDK7 inhibition. In summary, our findings highlight potential important roles of NEAT1_2 splicing and CDK7 in HSV-1 and/or IAV infection.

## Materials and methods

### RNA-seq data

All RNA-seq data analyzed in this study were downloaded from the SRA. SRA project IDs: HSV-1 WT infection time-course: SRP044766; HSV-1 Δ*vhs* infection time-course: SRP192356; RNA-seq of subcellular fractions: SRP110623, SRP189489, SRP191795 (same experiment but data for mutant viruses submitted separately); total RNA-seq for mock and WT KOS infection with ERCC spike-ins: SRP321121 (samples: SRR14632002, SRR14631995); T-HFs-ICP22 and T-HFs-ICP22/ICP27 cells, WT strain F and ΔICP22 infection: SRP340110; WT KOS, ΔICP27 and ICP27 overexpression: SRP189262; IAV infection time-courses: SRP091886, SRP103821; platelets: ERP003815; erythrocytes: SRP050333; CDK7 inhibition/knockdown: VCaP, LNCaP and DU145 cells: SRP179971; HCV-29 cells: SRP217721; TE7 and KYSE510 cells: SRP068450; C666-1, HK1 and HNE1 cells: SRP101458; BxPC3, MiaPaCa-2 and PANC1 cells: SRP165924; UM-Chor1 and CH22 cells: SRP166943, SRP270819; Nalm6 cells: SRP307127; DRB treatment of HEK293 cells: SRP055770. RNase R treatment: SRP197110, SRP152310.

### Linear RNA-seq read mapping

Sequencing reads were downloaded from SRA using the sratoolkit version 2.10.8 and aligned against the human genome (GRCh37/hg19) and human rRNA sequences using ContextMap2 version 2.7.9 [105] using BWA as short read aligner [50] and allowing a maximum indel size of 3 and at most 5 mismatches. For HSV-1 infection RNA-seq data, alignment also included the HSV-1 genome (Human herpesvirus 1 strain 17, GenBank accession code: JN555585). For the two repeat regions in the HSV-1 genome, only one copy was retained each, excluding nucleotides 1–9,213 and 145,590–152,222 from the alignment. SAM output files were converted to BAM files using samtools [106]. Following read mapping, fragment size distribution was determined from BAM files using the Picard CollectInsertSizeMetrics tool [107]. The linear RNA-seq mapping was used for identification of linear splice sites, filtering of identified circRNA reads (see below) and visualization of read coverage on genes.

### circRNA *de novo* detection

For *de novo* detection of circRNAs, we applied CIRI2 [45] and circRNA_finder [44] in parallel to all reads for each sample (including both reads aligned and unaligned in the linear RNA-seq mapping) (outline in Fig 2A). Combination of two complementary circRNA detection algorithms, such as CIRI2 and circRNA_finder, has been recommended to remove algorithm-specific false positives commonly observed in circRNA detection [43]. Both CIRI2 and circRNA_finder allow *de novo* detection of circRNAs using splice sites not annotated in the human genome. CIRI2 is based on identifying so-called paired chiastic clipping (PCC) signals in BWA alignments, which are pairs of clipped read alignments (i.e., local alignments for different substrings of the read) where a downstream part of the read aligns upstream of an upstream part of the read. Such reads are denoted as back-spliced junction (BSJ) reads and are then further filtered by CIRI2 to remove false positives (see Fig 2A and the original publication for details). For this purpose, we first aligned all reads using BWA with parameters

recommended by the developers of CIRI2 (= default BWA parameters, except for the -T option (= minimum score to output an alignment), which was set to 19). With default BWA options, matches in alignments are given a score of 1, mismatches a penalty of 4, the gap open penalty was 6 and the gap extend penalty was 1. As a consequence, mismatches on read segments flanking each side of the circular junction are only allowed if corresponding read segments are ≥ 24 nt.

circRNA_finder is based on filtering chimeric alignments obtained with STAR [49], in which different parts of a read align to two distinct regions of the genome in a manner not consistent with "normal" linear transcripts. We thus first aligned all reads with STAR using the default parameters used by the developers of circRNA_finder, i.e., chimSegmentMin (minimum total length of the chimeric segment) = 20, chimScoreMin (minimum total score of the chimeric segments) = 1, alignIntronMax (maximum intron length) = 100000, outFilterMismatchNmax (maximum number of mismatches per pair) = 4, alignTranscriptsPerReadNmax (maximum number of different alignments per read to consider) = 100000, outFilterMultimapNmax (maximum number of loci the read is allowed to map to) = 2. circRNA_finder then further filters chimeric alignments to identify circRNAs and remove false positives (see Fig 2A and the original publication for details).

CircRNAs were only further analyzed if they were detected independently by both algorithms. Circular reads identified by either algorithm were then pooled and subsequently all reads were removed that could be mapped in a linear fashion to the genome using ContextMap2. Only circRNAs with at least two supporting reads remaining in the same sample were further analyzed. Fragment lengths were not explicitly considered for circRNA detection, but all used algorithms have different criteria regarding the allowed distances and positions of the two reads in a read pair relative to each other (see original publications for details). The complete pipeline is available as a workflow for the workflow management system Watchdog [108] in the Watchdog workflow repository (https://github.com/watchdog-wms/watchdog-wms-workflows/tree/master/circRNA_Detection).

## Alignment-based circRNA detection

Candidate circular junction sequences were generated from putative spliced circRNAs sequences downloaded from circBase [48]. RNA-seq reads were aligned against circular junction sequences using BWA [50] with default parameters (i.e., -T is set to 30). BWA clips a read, i.e., outputs only a local alignment of the read, if the best local alignment score of the read minus a clipping penalty (default = 5) is at least as good as the best global alignment score. Thus, if a read only aligns on one side of the circular junction, only a local alignment to that side of the junction will be generated by BWA with at most a very short overlap to the other side of the junction with few mismatches. Reads were retained if they aligned to at least X nt (default X = 10) on either side of the junction and could not be mapped in a linear fashion anywhere to the genome using ContextMap2. In case of paired-end sequencing, the second read in the pair also had to align consistently within the circRNA region. circRNAs resulting from repetitive regions were discarded. The complete pipeline is outlined in S4 Fig.

## Identification of confirming read pairs

ContextMap2 is also capable of aligning read pairs, where each individual read can be aligned linearly to the genome, but the two reads are aligned in the wrong orientation relative to each other (= crosswise, see S14C Fig). Such crosswise alignments of read pairs are inconsistent with linear transcripts and were considered a confirming read pair for a circRNA if (i) both reads were aligned within the circRNA region with a genomic distance implying a fragment

size $\geq$ 500 nt if they originated from a linear transcript and (ii) the fragment size would be smaller if they originated from the circRNA. This fragment size cutoff was based on the observed distribution of fragment sizes, which showed that <3.7% of fragments exhibited a fragment size $\geq$ 500 nt.

## Read-through calculation

Number of read fragments per gene and in 5kb windows downstream of genes were determined from read alignments using featureCounts [109] and gene annotations from Ensembl (version 87 for GRCh37/hg19) [110]. For strand-specific RNA-seq, the stranded mode of featureCounts was used. All fragments (read pairs for paired-end sequencing or reads for single-end sequencing) overlapping exonic regions on the corresponding strand by $\geq$25 nt were counted for the corresponding gene. Downstream transcription for a gene was calculated as previously described [22] as the FPKM (Fragments Per Kilobase Million) in the 5kb windows downstream of genes divided by the gene FPKM multiplied by 100. %Read-through transcription was quantified as the difference in downstream transcription between infected/treated and uninfected/control cells, with negative values set to zero.

## Other bioinformatics analyses

Sashimi plots were generated with ggsashimi [111]. Heatmaps were generated with the heatmap.2 and pheatmap functions in R. Differential exon usage analysis was performed with the R Bioconductor package DEXSeq [112]. Log2 fold-changes in gene expression for WT and Δ*vhs* infection time-courses were obtained from our recent study [18]. In this study, we used DESeq2 [113] to compare gene expression between mock and each time-point of infection for 4,162 genes without read-in transcription originating from disrupted transcription termination for an upstream gene. Genes with read-in transcription were excluded from the analysis as read-in transcription can be mistaken for induction of gene expression.

## Supporting information

**S1 Fig. Statistics on identified circRNAs in HSV-1 WT and Δ*vhs* infection. (a, b)** Number of circRNAs identified for each sample of **(a)** total RNA and **(b)** 4sU-RNA with $\geq$ 2 reads per sample. **(c)** Scatterplot comparing the number of reads mapped to the human genome against the number of distinct circular splice sites identified by the *de novo* circRNA detection approach outlined in Fig 2A for the total RNA and 4sU-RNA time-courses in HSV-1 WT and Δ*vhs* infection. Numbers are shown separately for each replicate. **(d, e)** Percentage spliced host reads (= no. reads aligning to known splice junctions of host protein-coding genes / no. of mapped host reads × 100) for both replicates of total RNA-seq for **(d)** WT and **(e)** Δ*vhs* infection.
(PDF)

**S2 Fig. Enrichment of circRNAs in WT but not Δ*vhs* infection.** Scatterplots comparing normalized circRNA counts (normalized to the number of linear junction reads mapped to the host genome) between mock and 2, 4 and 6 h p.i. WT infection **(a-c)** and mock and 2, 4 and 6 h p.i. Δ*vhs* infection **(d-f)**. Linear regression analysis across all circRNAs (red line) was used to estimate the enrichment of circRNAs relative to linear mRNAs in HSV-1 infection compared to mock infection. The regression estimate for the enrichment is shown on the bottom right. The gray line indicates the diagonal, i.e., equal values on the x- and y-axis. The five most highly expressed circRNAs are marked by name. Corresponding scatterplots for 8 h p.i. are shown in

.
(PDF)

**S3 Fig. Results for circRNA normalization by total number of mapped reads.** Scatterplots comparing normalized circRNA counts normalized to the total number of reads mapped to the host genome between mock and 2, 4, 6 and 8 h p.i. WT infection **(a-d)** and mock and 2, 4, 6 and 8 h p.i. Δ*vhs* infection **(e-h)**. Linear regression analysis across all circRNAs (red line) was used to estimate the enrichment of circRNAs relative to all host reads in HSV-1 infection compared to mock infection. The regression estimate for the enrichment is shown on the bottom right. The gray line indicates the diagonal, i.e., equal values on the x- and y-axis. The five most highly expressed circRNAs are marked by name.
(PDF)

**S4 Fig. Outline of our alignment-based pipeline for circRNA detection.** For details see Materials and Methods. This approach allows increasing or decreasing the required overlap of a read with either side of a circular junction (=: X).
(PDF)

**S5 Fig. Results for trimmed reads.** Scatterplots comparing normalized circRNA counts (normalized to the number of linear junction reads **(a-d)** or to the total number of reads mapped to the host genome **(e-f)**) between mock and 2, 4, 6 and 8 h p.i. WT infection after trimming reads down to 76 nt. Linear regression analysis across all circRNAs (red line) was used to estimate the enrichment of circRNAs relative to linear mRNAs in HSV-1 infection compared to mock infection. The regression estimate for the enrichment is shown on the bottom right. The gray line indicates the diagonal, i.e., equal values on the x- and y-axis. The five most highly expressed circRNAs are marked by name.
(PDF)

**S6 Fig. Results for the alignment-based circRNA detection pipeline.** Scatterplots comparing normalized circRNA counts (obtained with the alignment-based circRNA detection pipeline outlined in S4 Fig and normalized to the number of linear junction reads mapped to the host genome) between mock and 2, 4, 6 and 8 h p.i. WT infection **(a-d)** and between mock and 2, 4, 6 and 8 h p.i. Δ*vhs* infection **(e-h)**. Linear regression analysis across all circRNAs (red line) was used to estimate the enrichment of circRNAs relative to linear mRNAs in HSV-1 infection compared to mock infection. The regression estimate for the enrichment is shown on the bottom right. The gray line indicates the diagonal, i.e., equal values on the x- and y-axis. The five most highly expressed circRNAs are marked by name.
(PDF)

**S7 Fig. Results for an independent total RNA-seq experiment including ΔICP27 infection.** Scatterplots comparing normalized circRNA counts (normalized to the number of linear junction reads mapped to the host genome) between mock and **(a)** WT, **(b)** ΔICP27 and **(c)** Δ*vhs* infection. CircRNA read counts were obtained with the alignment-based circRNA detection pipeline outlined in S4 Fig. Linear regression analysis across all circRNAs (red line) was used to estimate the enrichment of circRNAs relative to linear mRNAs in HSV-1 infection compared to mock infection. The regression estimate for the enrichment is shown on the bottom right. The gray line indicates the diagonal, i.e., equal values on the x- and y-axis. The five most highly expressed circRNAs are marked by name.
(PDF)

**S8 Fig. Enrichment of circRNAs in HSV-1 infection is due to loss of linear RNAs rather than increased circRNA synthesis. (a-d)** Scatterplots comparing circRNA counts obtained

for mock and WT KOS infection from the study of Dremel *et al.* [53] normalized by either **(a)** the number of linear junction reads mapped to the host genome, or **(b, c)** the number of reads mapping to ERCC spike-in sequences. CircRNA read counts were obtained with the alignment-based circRNA detection pipeline outlined in S4 Fig. **(c)** shows the same data as **(b)** with axes restricted to the range of 0 to 20. Linear regression analysis across all circRNAs (red line) was used to estimate in **(a)** the enrichment of circRNAs relative to linear mRNAs in HSV-1 infection compared to mock infection and in **(b, c)** the change in absolute circRNA abundances. The regression estimate is shown on the bottom right. The gray line indicates the diagonal, i.e., equal values on the x- and y-axis. The five most highly expressed circRNAs are marked by name. Normalization to ERCC spike-in read counts shows no absolute increase for most circRNAs in HSV-1 infection compared to mock infection, indicating that circRNAs are enriched relative to linear transcripts due to a general reduction of linear mRNAs during HSV-1 infection.
(PDF)

**S9 Fig. Impact of circRNA enrichment on differential gene and exon analyses. (a)** Boxplots showing the distribution of log2 fold-changes in gene expression between mock and 2, 4, 6 and 8 p.i. total RNA in WT and Δ*vhs* infection for genes containing at least one circRNA (blue) or no circRNA (gray). log2 fold-changes were taken from our recent study [18] (see also methods). Wilcoxon rank sum tests were used to assess whether log2 fold-changes were significantly increased for circRNA genes compared to other genes at each time-point of infection (p-values shown in red at the bottom). **(b)** Boxplots showing the distribution of normalized circRNA counts for circRNAs with (red) and without (blue) differential exon usage for exons within the circRNA region at each time-point of WT infection. **(c)** Percentage of expressed circRNAs (= circRNA count >0 in uninfected cells) for which at least one exon within the genomic region of the circRNA shows differential exon usage for the corresponding gene (determined with DEXSeq, multiple testing adjusted p-value ≤ 0.005) for each time-point of Δ*vhs* infection. **(d)** Boxplots showing the distribution of log2 fold-changes for exons located within circRNAs. For each circRNA, only the exon with the maximum absolute log2 fold-change is shown.
(PDF)

**S10 Fig. Enrichment of the hsa_circ_0003812 NEAT1_2 circRNA in total and 4sU-RNA and subcellular fractions in HSV-1 and mutant infections.** Enrichment (: = ε) compared to mock infection was calculated as the ratio of normalized hsa_circ_0003812 circRNA counts between infected and uninfected cells. Since no or very few reads were obtained for the hsa_circ_0003812 NEAT1_2 circRNA in uninfected cells, a pseudocount of 0.1 was used to avoid division by zero. Numbers on top indicate the number of circRNAs identified in the corresponding condition with a higher normalized circRNA count than hsa_circ_0003812. NA indicates that no hsa_circ_0003812 circular junction reads were not found in the particular condition.
(PDF)

**S11 Fig. Scatterplot comparing the number of reads mapped to the human genome against the number of distinct circular splice sites for chromatin-associated, nucleoplasmic, cytosolic and total RNA from the subcellular fractions experiment.**
(PDF)

**S12 Fig. Partial resistance of NEAT1_2 against RNase R treatment.** Sashimi plots show NEAT1 read coverage (overlay of replicates) and circular (red) and linear (same color as read coverage) splice junctions as arcs connecting acceptor and donor splice site in RNA-seq data

from the studies by **(a)** Xiao and Wilusz [20] and **(b)** Ungerleider et al. [55] (for details see below). Number of junction reads are annotated to arcs separately for replicates. Junctions are only shown if at least 3 and 4 reads, respectively, align by at least 10 nt on both sides of the junction. Genomic coordinates of NEAT1 transcripts and the HSV-1-induced circRNA are shown at the bottom. **(a)** Xiao and Wilusz performed RNA-seq for HeLa cells with and without (= Ctrl) RNase R treatment using two different protocols. The first protocol is the standard approach used for enriching circRNAs and employs RNase R treatment with a KCl-containing buffer for 15 min (2 biological replicates). In the second protocol, RNA was treated by E-PAP followed by digestion with RNase R in a LiCl-containing buffer (3 biological replicates). **(b)** Ungerleider et al. performed RNase R treatment for total, cytoplasmic and nuclear RNA obtained from Akata cells without (= Ctrl) and with induction of EBV activation. **(a, b)** NEAT1_2 is partially resistant to RNase R treatment due to the stabilizing triple helical structure at its 3'end, thus even after RNase R treatment reads from the linear NEAT1_2 transcript, in particular its 3'end, are obtained for total RNA from HeLa cells and nuclear RNA from Akata cells. For total and cytoplasmic RNA from Akata cells (both induced and noninduced), depletion of the linear NEAT1_2 transcript was successful.
(PDF)

**S13 Fig. Sashimi plots for total RNA time-courses of WT and Δ*vhs* infection.** For an explanation of sashimi plots see caption to Fig 3. Circular splice junctions are marked in red, linear splice junctions in the same color as read coverage.
(PDF)

**S14 Fig. Validation of the hsa_circ_0003812 NEAT circRNA. (a)** Heatmap of normalized circRNA counts (normalized to the number of linear junction reads mapped to the host genome) for all NEAT1_2 circRNAs in total and 4sU-RNA time-courses of WT infection identified with the alignment-based approach with a minimum overlap of at least 40 nt on either side of the circular junction. Columns represent individual circRNAs, which are ordered according to their genomic coordinates from the most 5' to the most 3'. The hsa_circ_0003812 NEAT1_2 circRNA is marked by a blue rectangle. **(b)** Read counts for circular junction reads for hsa_circ_0003812 identified with the alignment-based approach with a minimum overlap of at least 40 nt on either side of the circular junction (i.e., raw read counts used for calculating normalized circRNA counts for **(a)**) and read counts for confirming read pairs for hsa_-circ_0003812 identified with the approach outlined in **(c)**. Numbers are shown separately for both replicates of the total and 4sU RNA time-courses of WT infection. **(c)** Definition and detection of confirming read pairs for circRNAs. For further details see methods.
(PDF)

**S15 Fig.** Sashimi plots for 4sU-RNA time-courses of (a) WT and (b) Δ*vhs* infection and (c) nucleoplasmic and (d) total RNA from the subcellular fractions experiment for WT, Δ*vhs* and ΔICP27 infection. For an explanation of sashimi plots see caption to Fig 3. Circular splice junctions are marked in red, linear splice junctions in the same color as read coverage.
(PDF)

**S16 Fig.** Conservation of splice sites for (a) the hsa_circ_0003812 circular junction and (b-c) the two most frequent linear NEAT1_2 splice junctions (numbered (1) and (2) in Fig 3E). The left columns show the donor splice site (canonical GT signal at the intron 5' end) and the right column the acceptor splice site (canonical AG at the intron 3'end), with the red vertical lines marking the exon-intron boundaries. Please note that for the circular junction in **(a)**, the donor splice site is downstream of the acceptor splice site on the genome. Genome alignments

against the human genome (hg19) were obtained from the UCSC genome browser.
(PDF)

**S17 Fig. Sashimi plots for mock, WT-F and ΔICP22 infection at 8 h p.i.** For an explanation of sashimi plots see caption to Fig 3. Circular splice junctions are marked in red, linear splice junctions in the same color as read coverage.
(PDF)

**S18 Fig. Normalized circRNA counts for the hsa_circ_0003812 NEAT1_2 circRNA in IAV infection (cyan) and time-matched controls (gray).** Results are shown separately for H1N1, H3N2 and H5N1 infection and HTBE and MDM cells.
(PDF)

**S19 Fig. Enrichment of circRNAs in IAV infection. (a-g)** Scatterplots comparing normalized circRNA counts (normalized to the number of linear junction reads mapped to the host genome) for IAV infections at 18 h p.i. against time-matched controls. Linear regression analysis across all circRNAs (red line) was used to estimate the enrichment of circRNAs relative to linear mRNAs in IAV infection compared to time-matched controls. The regression estimate for the enrichment is shown on the bottom right. The gray line indicates the diagonal, i.e., equal values on the x- and y-axis. The most highly expressed circRNAs are marked by name. **(h)** Boxplots showing the distribution of log2 fold-changes between normalized circRNA counts in infection compared to time-matched controls for 7077 well-expressed circRNAs (normalized circRNA count >1 in at least one condition). A pseudocount of 0.1 was used to avoid division by zero. IAV strain and time-points of infection are color-coded (cyan = H1N1, green = H3N2, purple = H5N1, darker colors indicate later time-points). Values for the hsa_-circ_0003812 NEAT1_2 circRNA and the strongly enriched ZC3HAV1 circRNA are shown as blue squares and red triangles, respectively.
(PDF)

**S20 Fig. Boxplots showing distribution of % read-through in H1N1, H3N2 and H5N1 infection of HTBE and MDM cells.** Read-through was calculated as previously described in [22] (see also methods).
(PDF)

**S21 Fig. Results of the Snaptron search for NEAT1_2 splice junctions. (a)** Overview of number of projects and samples that were identified with the Snaptron search in recount3 for the four most frequent NEAT1_2 linear splice junctions induced in HSV-1 infection. The left-most column indicates regular expressions that were searched in the project and sample descriptions provided by recount3 using grep in R (case insensitive). The table shows only projects and samples for which at least one of the four most frequent NEAT1_2 linear splice junctions is covered by ≥ 10 reads. **(b)** Sashimi plots as in Fig 3 for two RNA-seq experiments (3 replicates each) in erythrocytes [70] and platelets [69]. For an explanation of sashimi plots, see caption to Fig 3. Circular splice junctions are marked in red, linear splice junctions in black.
(PDF)

**S22 Fig.** Sashimi plots showing read coverage and splice junctions upon (a) control, THZ1 or Act-D treatment in different concentrations in UM-Chor1 cells (2 replicates each) and (b) control or DRB treatment in HEK293 cells (2 replicates each). For an explanation of sashimi plots see caption to Fig 3. No circular splice junctions were found, thus only linear splice junctions are shown.
(PDF)

## Author Contributions

**Conceptualization:** Caroline C. Friedel.

**Formal analysis:** Marie-Sophie Friedl, Caroline C. Friedel.

**Funding acquisition:** Lars Dölken, Caroline C. Friedel.

**Investigation:** Marie-Sophie Friedl, Lara Djakovic, Thomas Hennig, Adam W. Whisnant, Lars Dölken, Caroline C. Friedel.

**Methodology:** Marie-Sophie Friedl, Caroline C. Friedel.

**Resources:** Lara Djakovic, Thomas Hennig, Adam W. Whisnant, Lars Dölken.

**Software:** Marie-Sophie Friedl, Michael Kluge, Caroline C. Friedel.

**Supervision:** Caroline C. Friedel.

**Visualization:** Marie-Sophie Friedl, Michael Kluge, Caroline C. Friedel.

**Writing – original draft:** Caroline C. Friedel.

**Writing – review & editing:** Thomas Hennig, Simone Backes, Lars Dölken, Caroline C. Friedel.

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
