## [Decision Letter · Decision Letter 0]

15 Aug 2022

PONE-D-22-19112HSV-1 and influenza infection induce linear and circular splicing of the long NEAT1 isoformPLOS ONE

Dear Dr. Friedel,

Thank you for submitting your manuscript to PLOS ONE. After careful consideration, we feel that it has merit but does not fully meet PLOS ONE’s publication criteria as it currently stands. Therefore, we invite you to submit a revised version of the manuscript that addresses the points raised during the review process.

All three reviewers made helpful comments on how to improve the writing and data presentation in your study. There are also several requests for better explanations of how data was analysed and/or interpreted. I ask you to take each request seriously and implement suggestions as much as possible.

We look forward to receiving your revised manuscript.

Kind regards,

Thomas Preiss, PhD

Academic Editor

PLOS ONE

Journal Requirements:

"This work was supported by the Deutsche Forschungsgemeinschaft (DFG) (FR2938/10-1 to C.C.F. and LD1275/6-1 to L.D.) and by the European Research Council (ERC-2016-CoG 721016 – HERPES to L.D.)".  

 "This work was supported by the Deutsche Forschungsgemeinschaft (www.dfg.de) grants FR2938/10-1 to C.C.F. and LD1275/6-1 to L.D. and by the European Research Council (erc.europa.eu) grant ERC-2016-CoG 721016 – HERPES to L.D.  The funders had no role in study design, data collection and analysis, decision to publish, or preparation of the manuscript".

Reviewers' comments:

Reviewer's Responses to Questions

**Comments to the Author**

1. Is the manuscript technically sound, and do the data support the conclusions?

Reviewer #1: Partly

Reviewer #2: Yes

Reviewer #3: Yes

2. Has the statistical analysis been performed appropriately and rigorously? 

Reviewer #1: Yes

Reviewer #2: Yes

Reviewer #3: I Don't Know

3. Have the authors made all data underlying the findings in their manuscript fully available?

Reviewer #1: Yes

Reviewer #2: Yes

Reviewer #3: Yes

4. Is the manuscript presented in an intelligible fashion and written in standard English?

Reviewer #1: Yes

Reviewer #2: Yes

Reviewer #3: Yes

5. Review Comments to the Author

Reviewer #1: In manuscript entitled “HSV-1 and influenza infection induce linear and circular splicing of the long NEAT1 isoform” authors combine multiple bioinformatics approaches to highlights potential role of NEAT1_2 in splicing and CDK7 in HSV-1 and/or IAV infection. Although the project design and manuscript preparations generally (see major point below) followed coherence with existing literature and adds valid contributions to the field, the manuscripts requires significant revisions and polishing in order to be published.

The major concern is to provide a more clear-cut definition of the observed alterations. The authors refer to “increase” or “decrease” or “induce” in certain circRNAs and common linear RNAs. Only from the design of the method, which is vaguely described, it may be assumed they are referring to relative alterations in library-size-normalised comparisons. Is this so? If yes, it need to be made very clear, as not all claims made in the manuscript would be supported by these data. Library-normalised alterations are not at all indicative neither of extent nor direction of absolute (actual concentration/abundance) alterations. Further, when assessed during events that trigger global transcriptme re-shaping, comparisons between conditions with regular approaches based on the assumption that only a small fraction of the transcriptome changes become unreliable or invalid. In essence, I think the authors need to provide additional/orthogonal proof to support the claims, such as FISH detection of the circRNA in situ, or its absolute quantification by other meas. The alternative is to substantially downplay the claims.

If I understand the statement in lines 154-157, the circRNA reads were normalised to the read counts of the genes producing them. All circRNA-producing genes or specific to the circRNA type? Remains unclear. In any case, this seems as a potentially dangerous approach as outlined in the paragraph above.

More minor points that are nonetheless important to address re summarised below.

Language needs edits and editing. Already in the abstract it is stated (lines 17-18) “vhs-mediated degradation of mRNAs leads to an accumulation of circular 18RNAs (circRNAs) relative to linear mRNAs”. Perhaps what is meant is “preferential accumulation of circRNAs versus linear RNA forms”?

Abstract needs to be re-written for clarity. It starts with virion host shut-off protein of herpes simplex virus 1 degradation effect on translated mRNAs, then drifts away to circRNAs in herpes simplex virus infection, then again away from virion host shut-off protein effects to the other immediate-early herpes simplex virus proteins. At this point, it already is confusing: is it circRNA production being elevated, or linear production suppressed, or there is degradation of linear RNA? It then goes on to influenza virus, which has completely different mechanisms, to show similar effects. Further, there is no clear segregation of results, hypotheses, assumptions and conclusions in the abstract. Perspectives and placement of the results on the global scope are also absent.

In Introduction, line 43 “CircRNAs form closed RNA-loops” – very confusing as overlaps with another broadly accepted term “RNA closed-loop” which is a loop and has nothing to do with a perfect covalent circle of circRNAs. Further, some studies that support the role of ICP22 and ICP27 on the splicing patterns of NEAT1_2 and/or NEAT1_1 could be mentioned. Nice mention of specific roles of all the proteins involved in the pathway. Furthermore, the authors have touched the roles of splicing of NEAT1_1 and NEAT1_2 splicing in HSV1, are the roles similar in influenza A? What does the previous literature state about this?

Not very clear in Methods how mapping and filtering was done. CircRNA detection relies on crossing the back-spliced junction in the approaches used. Thus, it is critical to understand how long were the insert sizes, how were thee treated and what happened to the reads with the 5 mismatches. How reliable was the remaining fully mapped overlap?

While supplementary figures are very numerous, I am not sure about the presentation approach and its logic. Are they necessary all? One does not need to include in the manuscript all what was plotted, just the bits that reinforce the conclusions made or hypotheses proposed.

Discussion is one of the better-written parts of the manuscript, although some parts are more of a fit to Results-type and placement in the global picture and leads to subsequent investigations could have been unfolded more extensively. Results section needs re-writing for the clarity and flow.

Among some others, word “strikingly” is used far too often.

Almost all figures use very small fonts and have font size disparity in the panels – very often making the information illegible.

Figure 1 in the introduction sections can be moved into the discussion or results sections as this figure contains relatively detailed results/cases, instead of illustrating the motivations behind the work. The authors may consider to include depiction of known role of cRNA in HSV-1, or potent role of NEAT1 and CDK7 in HSV-1, or even biogenesis of cRNA in relation to the specific circRNA in question.

A compact extract of material such as in Supplementary figure 1, 5 and 14 would be good to add in figure 1.

In Supplementary figure 18, it is best to use bars and not line plot – the measurement was not continuous.

Reviewer #2: The manuscript by Friedl et al investigates the effect of HSV1 infection on circular RNA formation using data previously generated by the group and uncovers an interesting effect of HSV1 on the linear and back splicing of the long NEAT1 isoform. The paper is clearly written, the study is methodologically thorough and the results are novel . I have a few comments and suggestions for improvement.

1. The circRNA detection pipeline uses a threshold of 2 reads/jusction for detecting circRNAs. Since sequencing depth varies considerably between samples (SFig2) – from 10 million to 100 million reads, this would strongly impact the number of circRNAs detected in each sample and likely explains the inter-sample variability. There also seems to be a bias in sequencing depth with the WT having less total reads than the �vhs in both total RNA and 4sU pulldown. The data in Fig1b (and related SuppFigs) should present numbers based on a threshold normalised to sequencing depth.

2. Line 144, the manuscript states “Consistent with previous reports [42] and the high stability of circRNAs, only a small number of circRNAs were detected in 4sU-RNA (119 and 194 in at least one sample of WT and Δvhs 4sU-seq, respectively).” I don’t think the results are consistent with ref42 which found ~ 1800 circRNAs at 4h of 4sU abeling. We also find a substantial number of circRNAs in 4sU labelled samples at 1-2h. So the low number observed here is either due to low sequencing depth or perhaps cell-type specific?

3. The columns of Fig2a need labelling. Also please add a diagram of start and end coordinates for all the NEAT1 circRNAs

4. To convincingly prove the circularization of the NEAT1_2 isoform the authors could provide Sanger sequencing data following PCR with inverted primers flanking the BSJ.

Minor

In Fig1b, SFig2a it would help to have the biological replicates next to each other.

InFig2b the track labelling is unclear. Two tracks have the same label “uninfected total” is one WT and one �vhs?

Reviewer #3: This bioinformatic analysis reveals the presence of linear splicing and back splicing of NEAT1, leading to circular RNA formation for this abundant nuclear transcript normally found within paraspeckles. In the study the authors have used publicly available RNA-seq datasets from different cell types, in either un-infected, infected with wildtype HSV or influenza viruses, or various mutant viral strains that lack key components. They have interrogated these datasets for circular RNA species, using many prediction tools and normalising in different ways. The main findings are a discovery of a circular RNA derived from the long isoform of NEAT1; the finding that this isoform is also extensively linearly spliced; and the finding that these species increase in abundance when cells are infected with either HSV, or influenza viruses.

I come at this review, not as a bioinformatics expert, but as a scholar with expertise in NEAT1. Given this expertise I can attest that the manuscript is thoughtful and has incorporated the existing NEAT1 literature to a thorough degree. In particular, I appreciate the discussion including paraspeckle protein RBP CLIP datasets into the circular and spliced RNA models; consideration for existing observations on transcriptional inhibitors and RNA stability of NEAT1 and paraspeckles; and understanding of the semi-extractability of NEAT1. In terms of scientific accuracy for the findings – these appear to be convincing (with respect to my limited bioinformatic capability), as (1) many datasets are used, from different cell types and with different viruses, all showing the same result of circular NEAT1 and (2) multiple prediction tools are used to isolate the circRNA from the datasets. Of course, it would be nice to have orthogonal methods confirm the results (wet lab experimentation etc) however, these are not essential.

The only thing I would like to see changed is inclusion of a new dedicated paragraph in the discussion where the authors speculate about the effects of circRNA and linear NEAT1 splicing on the actual appearance of paraspeckles within cells. This is something that needs to be considered as some of the datasets where these spliced species are found are linked to prior data suggesting paraspeckles increase (for example with viral infection) and other datasets linked to a decrease in paraspeckles/paraspeckle disappearance (eg. DRB and Actinomycin D). Yet in both instances the spliced/backspliced forms of NEAT1 are apparent. Hence it would be informative to arrive at a unified model of how the spliced NEAT1 relates to paraspeckles as seen down the microscope.

6. PLOS authors have the option to publish the peer review history of their article (what does this mean?). If published, this will include your full peer review and any attached files.

Reviewer #1: No

Reviewer #2: **Yes: **Irina Voineagu

Reviewer #3: No

---

## [Author Response · Author response to Decision Letter 0]

1 Sep 2022

The response to reviewers was uploaded as a separate PDF file.

---

## [Decision Letter · Decision Letter 1]

7 Oct 2022

HSV-1 and influenza infection induce linear and circular splicing of the long NEAT1 isoform

PONE-D-22-19112R1

Dear Dr. Friedel,

We’re pleased to inform you that your manuscript has been judged scientifically suitable for publication and will be formally accepted for publication once it meets all outstanding technical requirements.

Kind regards,

Thomas Preiss, PhD

Academic Editor

PLOS ONE

Additional Editor Comments (optional):

Reviewers' comments:

Reviewer's Responses to Questions

**Comments to the Author**

1. If the authors have adequately addressed your comments raised in a previous round of review and you feel that this manuscript is now acceptable for publication, you may indicate that here to bypass the “Comments to the Author” section, enter your conflict of interest statement in the “Confidential to Editor” section, and submit your "Accept" recommendation.

Reviewer #1: All comments have been addressed

Reviewer #2: All comments have been addressed

Reviewer #3: All comments have been addressed

2. Is the manuscript technically sound, and do the data support the conclusions?

Reviewer #1: Yes

Reviewer #2: Yes

Reviewer #3: Yes

3. Has the statistical analysis been performed appropriately and rigorously? 

Reviewer #1: Yes

Reviewer #2: Yes

Reviewer #3: I Don't Know

4. Have the authors made all data underlying the findings in their manuscript fully available?

Reviewer #1: Yes

Reviewer #2: Yes

Reviewer #3: Yes

5. Is the manuscript presented in an intelligible fashion and written in standard English?

Reviewer #1: Yes

Reviewer #2: Yes

Reviewer #3: Yes

6. Review Comments to the Author

Reviewer #1: The authors have addressed points raised during the peer review and have improved the manuscript to the level that is acceptable for publication on PLOS ONE.

It has been nice to see schematic in Figure 1 added and mapping and circRNA read assignment methods included or expanded in detail.

Reviewer #2: All comments have been addressed, thank you!

Reviewer #3: My comments have been addressed. As I am not familiar with the library normalisation processes in such cases I will leave it to the other two reviewers to comment on those changes.

7. PLOS authors have the option to publish the peer review history of their article (what does this mean?). If published, this will include your full peer review and any attached files.

Reviewer #1: **Yes: **Nikolay Shirokikh

Reviewer #2: **Yes: **Irina Voineagu

Reviewer #3: No

---

## [Editor Report · Acceptance letter]

12 Oct 2022

PONE-D-22-19112R1 

HSV-1 and influenza infection induce linear and circular splicing of the long NEAT1 isoform 

Dear Dr. Friedel:

I'm pleased to inform you that your manuscript has been deemed suitable for publication in PLOS ONE. Congratulations! Your manuscript is now with our production department. 

Kind regards, 

on behalf of

Prof Thomas Preiss 

Academic Editor

PLOS ONE